# Effect of Dietary Salt Intake on Risk of Gastric Cancer: A Systematic Review and Meta-Analysis of Case-Control Studies

**DOI:** 10.3390/nu14204260

**Published:** 2022-10-12

**Authors:** Xiaomin Wu, Liling Chen, Junxia Cheng, Jing Qian, Zhongze Fang, Jing Wu

**Affiliations:** 1Department of Epidemiology and Biostatistics, School of Public Health, Tianjin Medical University, Tianjin 300070, China; 2National Center for Chronic and Noncommunicable Disease Control and Prevention, Chinese Center for Disease Control and Prevention, Beijing 100050, China; 3Department of Toxicology and Sanitary Chemistry, School of Public Health, Tianjin Medical University, Tianjin 300070, China; 4Institute of Chronic Non-Communicable Disease Control and Prevention, Chongqing Center for Disease Control and Prevention, Chongqing 400042, China; 5Department of Social Medicine, School of Health Management, China Medical University, Shenyang 110122, China

**Keywords:** salt, gastric cancer, case-control study, meta-analysis, prevention

## Abstract

Aim: The effect of dietary salt intake on the risk of gastric cancer is not clear. A meta-analysis was performed to estimate the association between dietary salt intake and the risk of gastric cancer. Methods: Three major databases were searched to retrieve case-control studies published in English before 1 July 2022. Random effects model analysis was used to obtain the pooled odds ratios (*OR*s) and 95% confidence intervals (*CI*s) of the association between dietary salt intake and risk of gastric cancer. Subgroup analyses were used to identify possible sources of heterogeneity. Results: Thirty-eight case-control studies were included in this meta-analysis (total population: *n* = 37,225). The pooled *OR*s showed a significantly positive association between high salt intake and gastric cancer compared with low salt intake (*OR* = 1.55, 95% *CI* (1.45, 1.64); *p* < 0.001). In subgroup meta-analysis for geographic region, estimation method for dietary salt intake and the source of controls, this association was not changed. Conclusion: Higher dietary salt intake increased the risk of gastric cancer. This study has implications for the prevention of gastric cancer.

## 1. Introduction

Gastric cancer has long been a major public health issue [1]. Although the incidence and mortality rates of gastric cancer have declined in recent decades, it remains one of the most common cancers and the leading cause of cancer deaths [1,2]. According to GLOBOCAN estimates of cancer incidence and mortality, there were more than 1 million gastric cancer cases in 2020, resulting in more than 768,793 deaths [2]. The rise of gastric cancer as a leading cause of death has sparked concern. A prominent strategy is to prevent or delay the onset of gastric cancer.

The World Cancer Research Fund (WCRF), London, UK and its affiliates, including the American Institute for Cancer Research (AICR), Washington, DC, USA, have suggested cancer-prevention behaviors such as a healthy diet [3]. Lifestyle factors, including diet, may have an impact on cancer risk over a lifetime [3,4]. High salt consumption is one of the leading risk factors for a variety of non-communicable diseases, including gastric cancer [5]. Furthermore, one study founded that a high salt intake may be a risk factor for the development of gastric adenocarcinoma [6]. The association may be explained by two important factors. (1) Salt irritates the stomach wall and strongly enhances and promotes chemical gastric carcinogenesis [6,7]. (2) Excess salt may promote gastric *Helicobacter pylori* (*H. pylori*) colonization in the stomach, which is a known risk factor for gastric cancer [8,9]. High dietary salt intake is also contributing to the global burden of gastric cancer [10,11]. High sodium intake accounts for many the gastric cancer cases [10]. A healthy diet and lifestyle are required. By implementing the optimal lifestyle for all populations, half of all gastric cancer events could be prevented by the year 2031 [3,12]. If action is taken as early as possible, better effects can be achieved.

Among previous studies, the association between high dietary salt intake and gastric cancer was investigated, but the conclusion was inconsistent [13,14,15,16,17,18]. This is partly caused by the absence of reliable methods for estimating dietary salt intake. Taste preference, a food frequency questionnaire (FFQ), dietary behaviors, and other methods are used to estimate dietary salt intake. Inconsistent results may be due to the inconsistency of the estimation methods.

Given this, we performed a meta-analysis based on current published case-control studies to provide scientific and theoretical evidence for gastric cancer prevention. The focus should be on modifiable factors addressed as early as possible, which could show high effectiveness in preventing gastric cancer at a low cost.

## 2. Methods

The design, implementation, analysis, and reporting of our meta-analysis were reported in accordance with the PRISMA statement.

### 2.1. Data Sources and Search Strategy

We systematically searched three literature databases, including PubMed, Web of Science, and Cochrane Library, for studies published up to 1 July 2022 in English. The following Mesh terms and combinations of words were used for the literature search: (‘stomach neoplasms’ [Mesh] OR ‘gastric neoplasms’ OR ‘stomach cancer’ OR ‘gastric cancer’) AND (‘sodium, dietary’ [Mesh] OR ‘salt-heavy diet’ OR ‘high salt diet’ OR ‘salty food’). The searches were unlimited by time up to 1 July 2022, but were limited to human studies.

### 2.2. Selection Criteria and Exclusion Criteria

The studies were selected if they met all of the following criteria: (1) being a case-control study; (2) total sample size over 100; (3) assessment of salty food intake, preference of salty food, use of table salt and relevant indexes as exposure; (4) the authors reported odds ratio (*OR*) estimates, including 95% confidence intervals (*CI*s), for different salt intake categories. The studies were excluded if they met any of the following criteria: (1) being duplicate publications; (2) not being relevant; (3) being systematic reviews, meta-analyses, meeting abstracts, letters, and dissertations without the relevant information; (4) not being case-control studies; (5) *OR* and 95% *CI* not be reported. Studies with larger sample sizes was chosen among duplicate publications from the same case-control study. Exclusion criteria were applied sequentially by first screening the titles and abstracts, and then the full text. Duplicate records were excluded before screening began. The flow chart of the selection of studies is shown in Figure 1.

### 2.3. Data Extraction and Quality Assessment

Two investigators (Xiaomin Wu and Liling Chen) independently conducted the literature search, reviewed the retrieved articles, and extracted detailed information from included articles.

Any disagreement about whether a study met the inclusion criteria was resolved by group discussions with the third investigator (Junxia Cheng). The following characteristics of the identified studies and respective populations were recorded: first author, year of publication, country, region, gender, age (years) (mean/range), sample size of participants, match or not, the source of controls, estimation methods for dietary salt intake, comparisons, and adjustment variables for each study. The estimation methods for dietary salt intake in the different studies were provided in terms of total dietary salt intake or in terms of preference for salty food, or both. For our analysis, we used the outcome provided for total dietary salt intake whenever possible. Furthermore, we extracted *OR* estimates with the greatest adjustment.

Quality assessment was performed according to the Newcastle–Ottawa scale for observational studies [19]. This scale assigns a maximum of nine points to each study: four for selection of participants, two for comparability between both groups, and three for assessment of exposure. A greater score was considered to be an indicator of better quality on a scale of 9.

### 2.4. Statistical Analysis

STATA/SE 16 for Windows was used to analyze the data. The *OR*s and 95% *CI*s were considered as the effect size for all studies in this meta-analysis. The value from each study and the corresponding standard error were transformed into their natural logarithms to stabilize the variances and normalize their distribution. The pooled *OR* with corresponding 95% *CI* was estimated using a random effect model, weighting for the inverse of the variance. Heterogeneity among the studies was estimated using the *I^2^* statistic, with values of 25%, 50%, and 75% representing low, moderate, and high degrees of heterogeneity, respectively, with a *p* value < 0.10 deemed to be significant. A forest plot was used to visualize the *OR*s and 95% *CI*s of the included studies. A funnel plot was used to visualize a potential publication bias and Egger’s linear regression test was used to measure the asymmetry of the funnel plot, with a *p* value < 0.10 deemed to be significant. The influence of a single study was examined by sensitivity analysis. Subgroup analyses were used to identify associations between the risk of gastric cancer and relevant study characteristics (region and estimation method for dietary salt intake) as possible sources of heterogeneity. All tests were two-tailed and statistical significance was defined as *p* < 0.05.

## 3. Results

### 3.1. Literature Search and Study Characteristics

The study selection process and results from the literature search are shown in Figure 1. Of a total of 1462 publications retrieved, 38 studies were identified that met the inclusion criteria. The relevant characteristics of the 38 studies included in the meta-analysis are reported in Table 1. Overall, the meta-analysis involved 37,225 participants from 20 countries (11 studies from China; 4 from Korea; 4 from Italy; 2 from Iran; 2 from Turkey; 1 from France, England, Spain, Japan; Puerto Rico, Sweden, Mexico, Thailand, Uruguay, Colombia, Portugal, Serbia, Canada, Ecuador, and Poland). In all studies; the dietary salt intake was estimated by FFQ or relevant tests. The estimation methods for dietary salt intake are shown in Table 2. The Newcastle–Ottawa Scale was used to assess the quality of included articles. The results of the quality scoring are shown in Table 1. A summary of the characteristics and quality assessment of the included studies is listed in Table 1 and Table 2. The information on the adjustment variables for each study is shown in Appendix A.

### 3.2. Effects of Dietary Salt Intake on Risk of Gastric Cancer

There were 38 case-control studies to evaluate the risk of dietary salt intake with gastric cancer. The form of a forest plot is used to show the results of the pooled analyses in Figure 2 (pooled *OR*: 1.55, 95% *CI*: 1.45–1.64). There was significant heterogeneity between studies (*p* < 0.001, *I^2^* = 82.8%). There was publication bias detected in the meta-analysis (*p* < 0.001) (The funnel plot is shown in the Appendix A). Additional analyses were performed to check for potential sources of heterogeneity that might explain the association between high dietary salt intake and gastric cancer events. Subgroup analyses were performed.

### 3.3. Sensitivity Analysis

We explored the effect of a single study on the pooled *OR* with the sensitivity analysis. The result indicated that the Pakseresht et al. study influenced the pooled *OR* [49]. If this study was omitted, the pooled *OR* would be 1.77 (95% *CI*: 1.65–1.90) (shown in Table 3). The results showed that high dietary salt intake was significantly associated with a greater risk of gastric cancer compared with low salt intake after omitting single studies one by one.

### 3.4. Subgroup Analyses by Region, Estimation Methods for Dietary Salt Intake and the Source of Controls

The relationship between high dietary salt intake and risk of gastric cancer was not significantly different between geographic regions, estimation methods of dietary salt intake, and the source of controls (shown in Figure 3, Figure 4 and Figure 5). The pooled *OR*s were changed after stratifying by geographic region. The pooled *OR*s of gastric cancer for the salt intake were 1.71 (95% *CI*, [1.51, 1.95]) for studies conducted in Europe, 1.48 (95% *CI*, [1.37, 1.59]) for studies conducted in Asia, and 1.65 (95% *CI*, [1.38, 1.97]) for studies conducted in America; there was statistically significant heterogeneity among studies of salt intake in Europe (*p* < 0.001 and *I^2^* = 77.2%), Asia (*p* < 0.001 and *I^2^* = 86.3%), and America (*p* = 0.006 and *I^2^* = 72.1%) (shown in Figure 3). Furthermore, stratifying by estimation method for dietary salt intake, the pooled *OR*s of gastric cancer for salt intake were 1.38 (95% *CI*, [1.29, 1.49]) for studies that estimated salt addition and 2.03 (95% *CI*, [1.81, 2.27]) for studies that estimated consumption of salty foods or salt preference; there was statistically significant heterogeneity among studies that estimated salt addition (*p* < 0.001 and *I^2^* = 87.2%) and there was statistically medium heterogeneity among studies that estimated consumption of salty foods or salt preference (*p* < 0.001 and *I^2^* = 66.0%) (shown in Figure 4). The pooled *OR*s of gastric cancer for salt intake were 1.39 (95% *CI*, [1.30, 1.49]) for studies with controls from the community and 2.19 (95% *CI*, [1.93, 2.49]) for studies with controls from hospitals; there was statistically significant heterogeneity in studies (*I^2^* = 77.9% for population-based studies and *I^2^* = 81.7% for hospital-based studies, *p* < 0.001) (shown in Figure 5).

## 4. Discussion

In this meta-analysis, it was found that (1) compared with low dietary salt intake, high dietary salt intake could increase the gastric cancer risk (overall *OR* = 1.55, 95% *CI* [1.45, 1.64]; *p* < 0.001). (2) In subgroup analyses by geographical region and estimation method for salt intake, the significantly positive association was not changed.

Our findings suggest that a high salt intake is associated with gastric cancer, which is consistent with the findings of other studies [13,14]. In addition, this association was not confirmed in some studies [15,34,38,49]. The different results may be explained by several important factors. (1) The frequency of salty food consumption was used to estimate salt intake in some studies, but the definition of salty foods was different. Soy sauce, which has been shown to have a protective effect against gastric cancer in other studies, was classified as a salty food in the Hyun Ja Kim et al. study [34]. (2) One study conducted in Poland estimated dietary salt intake through weekly frequency of salt consumption from food. However, foods that are of high salt content are universally consumed in Poland, so it was difficult to detect any differences [38]. (3) One study used a salt-added increment to estimate the association between high dietary salt intake and gastric cancer, but the incremented unit was 1 g [49]. The estimation method may underestimate the actual association [49]. (4) Researchers used the lowest intake as the reference group in their studies. However, in one study, the no opinion group was chosen as the reference group; this choice may not detect the actual effect of high dietary salt intake on gastric cancer [18]. (5) Confounding factors, comorbidities, and observational bias could all have an impact on the actual association.

Similar to other studies, our study also found that dietary salt intake is associated with gastric cancer. There are several mechanisms to explain this association: (1) The gastric mucosa could be damaged by high salt concentration directly, which leads to hyperplasia of the gastric pit epithelium and increases the probability of endogenous mutations [13,14]. Additionally, the damage to gastric mucosa could increase DNA damage and glandular atrophy [14]. (2) High salt intake could accelerate the procedure of intestinal metaplasia, which could develop into early gastric cancer [14]. (3) Salty foods that have too much nitrate and nitrite could contribute to the formation of N-nitroso compounds [53]. The carcinogenic effect of nitroso compounds may be promoted or enhanced by high salt intake [4]. Additionally, high salt intake may also promote or enhance the effect of other carcinogens [4]. (4) High salt intake increases *H. pylori* colonization in the stomach. *H. pylori* is one of the main predisposing factors for gastric cancer [4,13,14]. The cag pathogenicity island is one of the *H. pylori* virulence determinants, which could increase gastric cancer risk [54]. More severe gastric injury in the stomach was induced by cag-positive strains compared with cag-negative strains, and cag-positive strains further augment the risk for gastric cancer [54]. Elevated salt concentrations caused an upregulation of the cagA gene in some strains, enhancing cagA’s ability to translocate into gastric epithelial cells [54,55]. This indicates that high dietary salt intake could enhance the carcinogenic effects of cagA+ *H. pylori* strains [14]. (5) High salt intake could alter the viscosity of the protective mucous barrier, disrupt immune homeostasis, and increase susceptibility to *H. pylori* infection [11,56,57]. These factors would result in chronic inflammation, such as atrophic gastritis and gastric ulcers both of which are common precancerous diseases [13,14,56,58].

There was significant heterogeneity among the included studies. This situation was also observed in other comparable studies [13,14]. The potential sources of heterogeneity were checked with further subgroup analyses, which might explain the association between dietary salt intake and gastric cancer events. Among the studies that estimated salt intake by consumption of salty foods or salt preference, the heterogeneity was decreased. This indicates that estimation methods for dietary salt intake may be a source of heterogeneity. It is difficult to quantify the intake of sodium, which is the main component of salt. The FFQ was used to estimate dietary salt intake in most studies. The actual intake of salt could not be estimated through the FFQ, and recall bias is inevitable. Cases tend to overestimate their exposure to risk factors; possibly, this may lead to a spurious association between risk factors and disease [59].

Publication bias existed in our meta-analysis. Negative results were not be reported, especially in studies published in the 1990s, which is the main source of publication bias.

### Limitations

There exist several potential limitations in this study. First, we only included studies published in English and we did not search grey literatures. The actual total number of eligible studies may be larger than the currently included studies. Second, confounding risk factors such as *H. pylori*, smoking, and other relevant risk factors were not able to be considered in this meta-analysis. Third, given the observational nature of the included studies, our study lacked evidence to clarify causation. Fourth, the estimation methods for dietary salt intake, which contributed to the heterogeneity of this study, were not classified in more detail.

## 5. Conclusions

In conclusion, it was indicated that higher dietary salt intake increased the risk of gastric cancer. Participants who prefer salty foods need to receive dietary education and diet management for the prevention of gastric cancer. This finding has important public health implications. Societies and individuals may succeed in lowering their risk for gastric cancer by reducing dietary salt intake. An additional meta-analysis that includes more cohort studies is needed.

## Figures and Tables

**Figure 1 nutrients-14-04260-f001:**
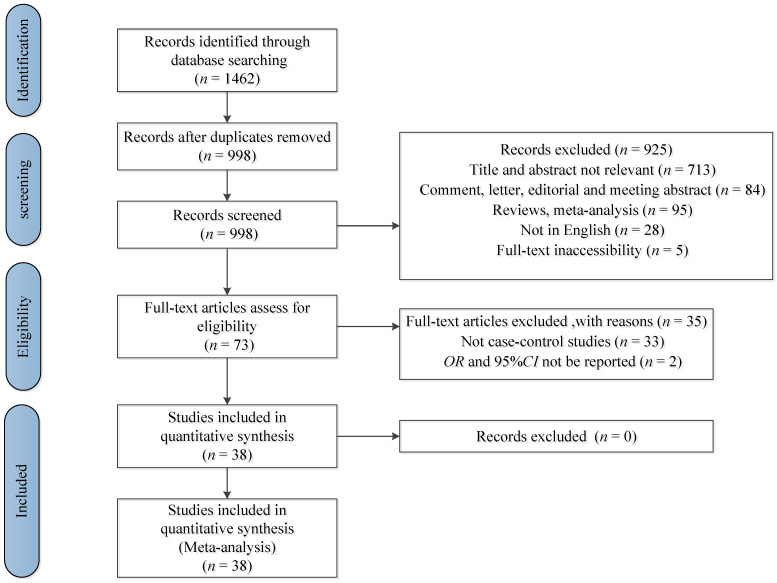
Flow chart of selection of studies for the meta-analysis.

**Figure 2 nutrients-14-04260-f002:**
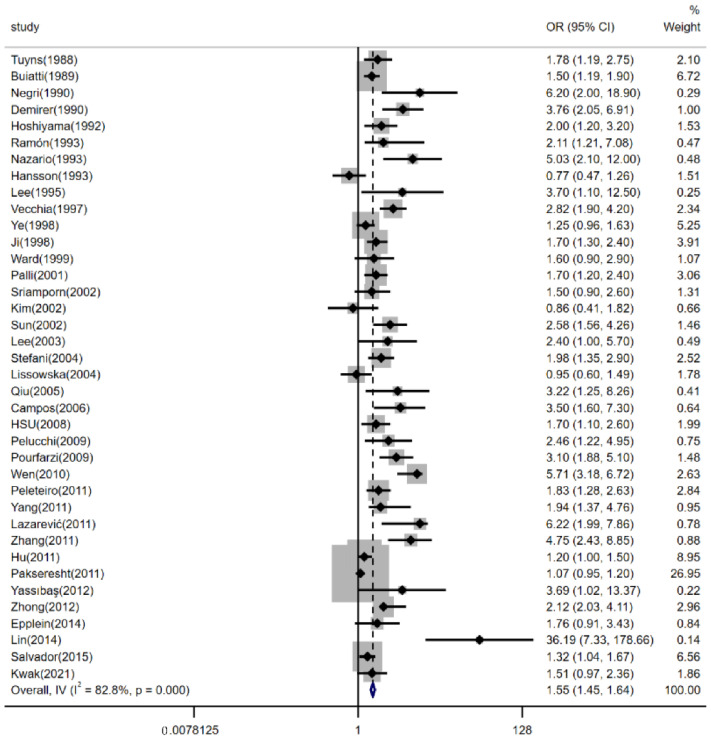
Forest plot of associations between high dietary salt intake and gastric cancer risk. References [9,15,16,17,18,20,21,22,23,24,25,26,27,28,29,30,31,32,33,34,35,36,37,38,39,40,41,42,43,44,45,46,47,48,49,50,51,52] are cited in the Figure.

**Figure 3 nutrients-14-04260-f003:**
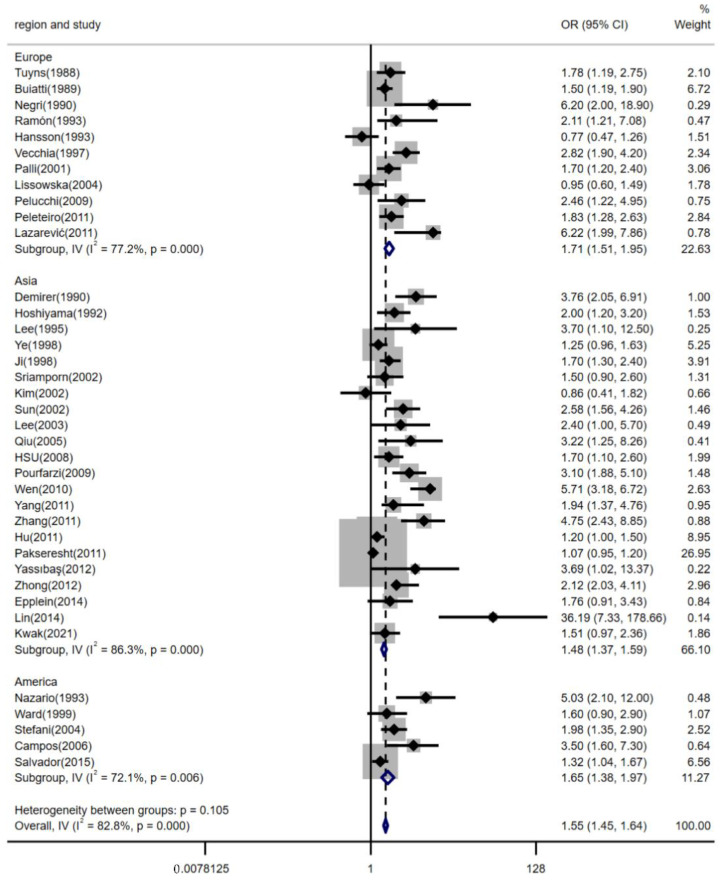
Forest plot of associations between high dietary salt intake and gastric cancer risk among different regions. References [9,15,16,17,18,20,21,22,23,24,25,26,27,28,29,30,31,32,33,34,35,36,37,38,39,40,41,42,43,44,45,46,47,48,49,50,51,52] are cited in the Figure.

**Figure 4 nutrients-14-04260-f004:**
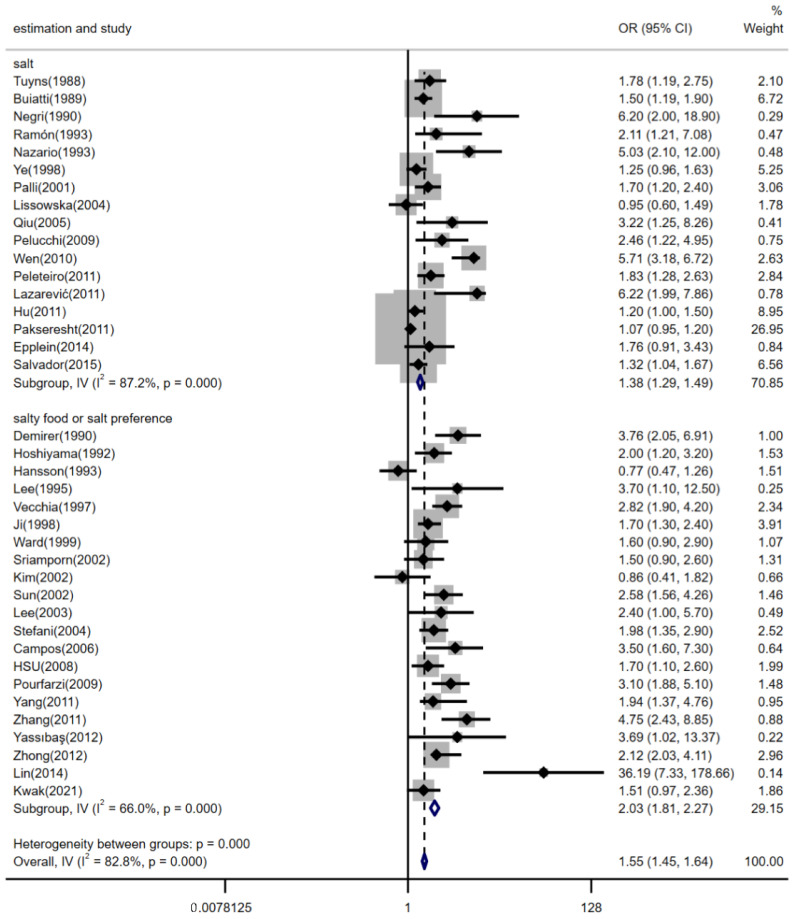
Forest plot of associations between high dietary salt intake and gastric cancer risk among different estimation methods for salt intake. References [9,15,16,17,18,20,21,22,23,24,25,26,27,28,29,30,31,32,33,34,35,36,37,38,39,40,41,42,43,44,45,46,47,48,49,50,51,52] are cited in the Figure.

**Figure 5 nutrients-14-04260-f005:**
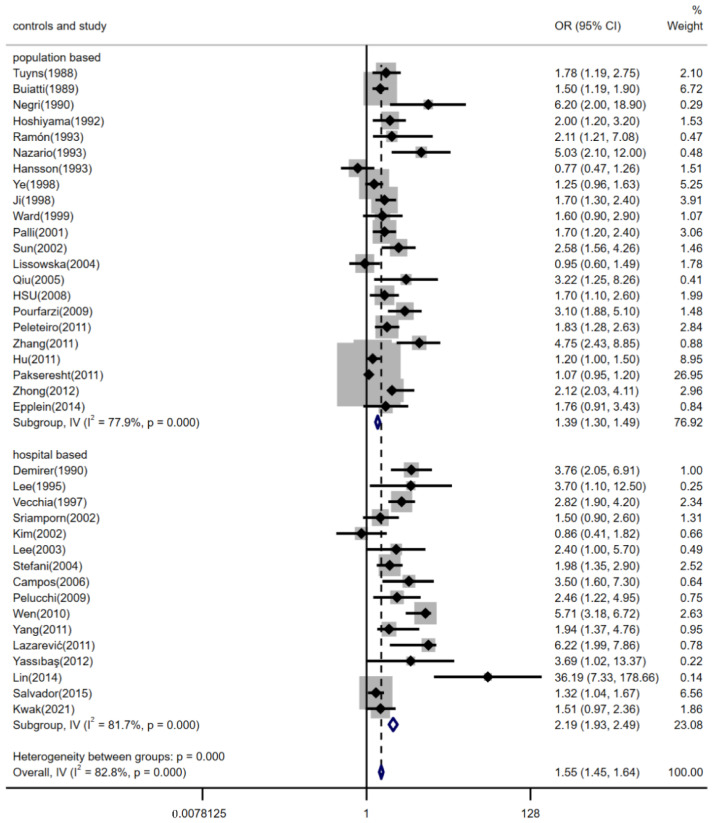
Forest plot of associations between high dietary salt intake and gastric cancer risk among the source of controls. References [9,15,16,17,18,20,21,22,23,24,25,26,27,28,29,30,31,32,33,34,35,36,37,38,39,40,41,42,43,44,45,46,47,48,49,50,51,52] are cited in the Figure.

**Table 1 nutrients-14-04260-t001:** Characteristics of the case-control studies included in the meta-analysis.

First Author	Year	Country	Region	Male (*n*)	Age (Years) Mean/Range		Sample Size	Quality Score
					Case	Control		
Tuyns [20]	1988	France	Europe	1597	—	—	4061	5
Buiatti [21]	1989	Italy	Europe	1345	≤75	≤75	2175	5
Negri [22]	1990	England	Europe	219	61	64	282	8
Demirer [23]	1990	Turkey	Asia	131	55	52	200	7
Hoshiyama [24]	1992	Japan	Asia	699	—	—	699	6
Ramón [25]	1993	Spain	Europe	297	62	61	351	8
Nazario [26]	1993	Puerto Rico	America	—	≥30	≥30	271	8
Hansson [27]	1993	Sweden	Europe	662	67.7	67.0	1017	8
Lee [28]	1995	Korea	Asia	264	>25	>25	426	7
Vecchia [29]	1997	Italy	Europe	1662	61	55	2799	5
Ye [15]	1998	China	Asia	699	30–78	30–78	816	8
Ji [30]	1998	China	Asia	1589	61	59	2575	8
Ward [31]	1999	Mexico	America	—	≥20	≥20	972	8
Palli [32]	2001	Italy	Europe	567	—	—	943	6
Sriamporn [33]	2002	Thailand	Asia	254	—	—	393	7
Kim [34]	2002	Korea	Asia	186	—	—	314	7
Sun [35]	2002	China	Asia	568	59.8	59.5	840	8
Lee [36]	2003	Korea	Asia	166	—	—	268	7
Stefani [37]	2004	Uruguay	America	840	30–89	30–89	1200	7
Lissowska [38]	2004	Poland	Europe	479	—	—	737	8
Qiu [39]	2005	China	Asia	176	63	60	176	6
Campos [40]	2006	Colombia	America	407	—	—	647	7
Hsu [41]	2008	China	Asia	131	66.0	51.8	349	8
Pelucchi [42]	2009	Italy	Europe	429	63	63	777	7
Pourfarzi [43]	2009	Iran	Asia	416	65.4	64.3	611	8
Wen [44]	2010	China	Asia	642	58.9	57.7	900	7
Peleteiro [16]	2011	Portugal	Europe	503	18–92	18–92	1071	8
Yang [45]	2011	China	Asia	642	52.1	52.4	900	7
Lazarević [46]	2011	Serbia	Europe	—	65.8	65.8	306	7
Zhang [47]	2011	China	Asia	424	53.3	52.8	645	6
Hu [48]	2011	Canada	America	1528	57.1	60.1	6221	6
Pakseresht [49]	2011	Iran	Asia	427	66.3	62.9	590	6
Yassıbaş [50]	2012	Turkey	Asia	132	57.4	57.9	212	7
Chen [9]	2012	China	Asia	390	53.1	52.8	617	6
Epplein [51]	2014	China	Asia	677	62.6	63.6	677	8
Lin [52]	2014	China	Asia	241	59.1	56.5	316	6
Salvador [17]	2015	Ecuador	America	95	62.0	55.5	257	7
Kwak [18]	2021	Korea	Asia	412	56.9	56.2	614	7

Note: “—” not reported or not acquired.

**Table 2 nutrients-14-04260-t002:** Detailed exposure of the studies.

First Author	Publication Year	Match	Source of Controls	Estimation Method for Dietary Salt Intake	Comparisons
Tuyns [20]	1988	Non-matched	Population-based	Addition of salt	Never
Buiatti [21]	1989	Non-matched	Population-based	Add salt	Never/seldom
Negri [22]	1990	Matched	Population-based	Levels of salt intake	Low
Demirer [23]	1990	Matched	Hospital-based	Consumption frequency of salted foods	No consumption/“rare” consumption/ Once or twice a month
Hoshiyama [24]	1992	Non-matched	Population-based	Preference for salty foods	Low
Ramón [25]	1993	Matched	Population-based	Salt intake	<1.96 (g/day)
Nazario [26]	1993	Non-matched	Population-based	Salt index	<6.979 (g/week)
Hansson [27]	1993	Matched	Population-based	Salted fish	Low
Lee [28]	1995	Non-matched	Hospital-based	Salt preference	Low
Vecchia [29]	1997	Non-matched	Hospital-based	Salt preference	Low
Ye [15]	1998	Matched	Population-based	Salt	≤0.25 kg/month
Ji [30]	1998	Matched	Population-based	Consume salted foods	Occasionally
Ward [31]	1999	Non-matched	Population-based	Salty snacks/crackers	Never
Palli [32]	2001	Non-matched	Population-based	Sodium intake	Low tertile
Sriamporn [33]	2002	Matched	Hospital-based	Salted food	Low
Kim [34]	2002	Matched	Hospital-based	Salted food	Low
Sun [35]	2002	Matched	Population-based	Salt preference	Moderate
Lee [36]	2003	Non-matched	Hospital-based	Salt fermented fish	<1/month
Stefani [37]	2004	Matched	Hospital-based	Salted meat consumption	Low
Lissowska [38]	2004	Matched	Population-based	Weekly frequency of salt consumption	Low
Qiu [39]	2005	Non-matched	Population-based	Daily intake of sodium	Low
Campos [40]	2006	Matched	Hospital-based	Salting meals before tasting	No
Hsu [41]	2008	Non-matched	Population-based	Salty food intake	Low
Pelucchi [42]	2009	Matched	Hospital-based	Intake of sodium	Low
Pourfarzi [43]	2009	Matched	Population-based	Salt preference	Not salty
Wen [44]	2010	Matched	Hospital-based	STST ≥ 5	STST < 5
Peleteiro [16]	2011	Non-matched	Population-based	Use of table salt (salt consumption by visual analogical scale)	<35 (mm)
Yang [45]	2011	Matched	Hospital-based	Salt taste preference	Not salty
Lazarević [46]	2011	Matched	Hospital-based	Intake of salt	Low
Zhang [47]	2011	Non-matched	Population-based	Salt taste preference *	0.9 (g/L)
Hu [48]	2011	Non-matched	Population-based	Added salt at table	Never
Pakseresht [49]	2011	Non-matched	Population-based	Salt	Per g
Yassıbaş [50]	2012	Matched	Hospital-based	Salt status of dishes	Salt-free
Chen [9]	2012	Non-matched	Population-based	Salt taste preference *	<1.8
Epplein [51]	2014	Matched	Population-based	Intake of sodium	Low
Lin [52]	2014	Matched	Hospital-based	Salt taste preference	Not salty
Salvador [17]	2015	Non-matched	Hospital-based	Adding salt >50% of meals	No
Kwak [18]	2021	Matched	Hospital-based	Salt taste preference	No opinion

Note: STST: Salt taste sensitivity threshold, *: the salt preference was assessed by threshold level of salty taste.

**Table 3 nutrients-14-04260-t003:** OR estimates and 95% CI after omitting studies one by one.

Study Omitted	*OR*	95% *CI*
Tuyns (1988) [20]	1.54	(1.45, 1.64)
Buiatti (1989) [21]	1.55	(1.45, 1.65)
Negri (1990) [22]	1.54	(1.45, 1.64)
Demirer (1990) [23]	1.53	(1.44, 1.63)
Hoshiyama (1992) [24]	1.54	(1.45, 1.64)
Ramón (1993) [25]	1.54	(1.45, 1.64)
Nazario (1993) [26]	1.54	(1.45, 1.63)
Hansson (1993) [27]	1.56	(1.47, 1.66)
Lee (1995) [28]	1.54	(1.45, 1.64)
Vecchia (1997) [29]	1.52	(1.43, 1.62)
Ye (1998) [15]	1.56	(1.47, 1.66)
Ji (1998) [30]	1.54	(1.45, 1.64)
Ward (1999) [31]	1.54	(1.45, 1.64)
Palli (2001) [32]	1.54	(1.45, 1.64)
Sriamporn (2002) [33]	1.55	(1.45, 1.64)
Kim (2002) [34]	1.55	(1.46, 1.65)
Sun (2002) [35]	1.53	(1.44, 1.63)
Lee (2003) [36]	1.54	(1.45, 1.64)
Stefani (2004) [37]	1.54	(1.44, 1.63)
Lissowska (2004) [38]	1.56	(1.47, 1.66)
Qiu (2005) [39]	1.54	(1.45, 1.64)
Campos (2006) [40]	1.54	(1.45, 1.63)
Hsu (2008) [41]	1.54	(1.45, 1.64)
Pelucchi (2009) [42]	1.54	(1.45, 1.64)
Pourfarzi(2009) [43]	1.53	(1.44, 1.63)
Wen (2010) [44]	1.50	(1.40, 1.59)
Peleteiro (2011) [16]	1.54	(1.45, 1.64)
Yang (2011) [45]	1.54	(1.45, 1.64)
Lazarević (2011) [46]	1.53	(1.44, 1.62)
Zhang (2011) [47]	1.53	(1.44, 1.63)
Hu (2011) [48]	1.58	(1.49, 1.69)
Pakseresht (2011) [49]	1.77	(1.65, 1.90)
Yassıbaş (2012) [50]	1.54	(1.45, 1.64)
Chen (2012) [9]	1.53	(1.44, 1.63)
Epplein (2014) [51]	1.54	(1.45, 1.64)
Lin (2014) [52]	1.54	(1.45, 1.63)
Salvador (2015) [17]	1.56	(1.47, 1.66)
Kwak (2021) [18]	1.55	(1.45, 1.64)

## Data Availability

All reported data are available in the manuscript.

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
