# Peer review of "Effect of Dietary Salt Intake on Risk of Gastric Cancer: A Systematic Review and Meta-Analysis of Case-Control Studies"

_nutrients, 2022, doi:10.3390/nu14204260_

Round 1
Reviewer 1 Report
Thank you for the opportunity to review the manuscript, “Effect of Dietary Salt Intake on Risk of Gastric Cancer: A Systematic Review and Meta-analysis of Case-control Studies,” submitted to nutrients. This manuscript evaluates regional case-control studies, and studies by their measurement of salt intake. It is consistent with numerous reports in the last 10 years of a small but significant association between dietary sodium intake and gastric cancer. It is somewhat unfortunate that confounding risk factors such as H. pylori, smoking, and other relevant risk factors were not able to be considered in this meta analysis. This point aside, the authors seem to have carefully evaluated case-control studies from across the globe, and provide further support for the association between dietary sodium intake and gastric cancer. They satisfactorily describe the few outliers they observed in the meta analysis. The methods seem appropriate, specifically the research and statistical methodologies are sufficiently documented.
Author Response
Dear professor,
Thank you for your letter and for the reviewers’ comments concerning our manuscript entitled “Effect of Dietary Salt Intake on Risk of Gastric Cancer: A Systematic Review and Meta-analysis of Case-control Studies” (manuscript ID: nutrients-1889898). Those comments are very valuable and helpful for revising and improving our paper, as well as the important guiding significance to our research. We have carefully revised our manuscript according to those comments, and uploaded our revised manuscript with all the changes highlighted by using the track changes mode. Besides, we provide this cover letter to explain the details of our revisions of the manuscript and our point-by-point responses to the reviewers’ comments. We also have uploaded the modified full text version. We hope that our revised manuscript is acceptable to publication. If you have any questions about our revised manuscript, please inform us without hesitation. Looking forward to hearing from you.
Sincerely,
Jing Wu, MD, PhD
Professor
National Center for Chronic and Non-Communicable Disease Control and Prevention Chinese Center for Disease Control and Prevention
Beijing 100050, China.
E-mail address: [email protected]
Zhongze Fang, MD, PhD
Professor
Department of Toxicology and Sanitary Chemistry
School of Public Health
Tianjin Medical University
Tianjin 300070, China
E-mail address: [email protected]
REVIEWER 1:
- Thank you for the opportunity to review the manuscript, “Effect of Dietary Salt Intake on Risk of Gastric Cancer: A Systematic Review and Meta-analysis of Case-control Studies,” submitted to nutrients. This manuscript evaluates regional case-control studies, and studies by their measurement of salt intake. It is consistent with numerous reports in the last 10 years of a small but significant association between dietary sodium intake and gastric cancer. It is somewhat unfortunate that confounding risk factors such as H. pylori, smoking, and other relevant risk factors were not able to be considered in this meta-analysis. This point aside, the authors seem to have carefully evaluated case-control studies from across the globe, and provide further support for the association between dietary sodium intake and gastric cancer. They satisfactorily describe the few outliers they observed in the meta-analysis. The methods seem appropriate, specifically the research and statistical methodologies are sufficiently documented.
Response: Many thanks for this comment. It's really a valuable advice. Unfortunately, the confounding risk factors such as H. pylori, smoking, and other relevant risk factors were not be considered in this meta-analysis. This is one of the limitations of this meta-analysis. We have supplemented it in discussion part, and copy it here for your check. We have already collected some relevant data and our cohort is still under observation. We will analyze the relevant data after thorough research. Hope you will continue to pay attention to our following study. Thank you.
There exist several potential limitations in this study. First, we only included studies published in English. And we did not search grey literatures. The actual total number of included studies maybe larger than current included studies. Second, confounding risk factors such as H. pylori, smoking, and other relevant risk factors were not able to be considered in this meta-analysis. Third, given the observational nature of the included studies, our study lacked evidence to clarify the causation. Fourth, the estimated methods of dietary salt intake, which contributed to the heterogeneity of this study, were not classified in more detail.
We tried our best to improve the manuscript and made some changes in the manuscript. These changes will not influence the content and framework of the paper. We appreciate the Editors/Reviewers’ warm work earnestly, and hope that the correction will meet with approval. Once again, thank you very much for your comments and suggestions.

Reviewer 2 Report
This work has the merit of paying attention to the development of gastric cancer which is still one of the 4 great killers all over the world. I would have preferred more clarity on the content and quality of the salty food (On the other hand I know the difficulties about different kind of food in different countries). this aspect could be useful to shed more light on salty foods most at risk
Author Response
Response: Thank you for your recommendation. It's really a valuable advice, we will consider your advice next. We will collect relevant data and analyze the relevant data after thorough research. Hope you will continue to pay attention to our following study. Thank you.
We appreciate the Editors/Reviewers’ warm work earnestly, and hope that the correction will meet with approval. Once again, thank you very much for your comments and suggestions.

Reviewer 3 Report
My main concerns regarding this manuscript include:
- The manner in which the systematic review portion of the manuscript was carried out - the authors did not exclude duplicate records correctly, then the authors did not apply exclusion criteria sequentially by screening first the titles and abstracts, and second the full text.
- Studying the association between dietary salt exposure and gastric cancer is subject to several methodological difficulties due to the limitations of the methods available to measure dietary salt exposure. The authors pooled together estimates that quantify the association between dietary salt exposure defined according to different criteria and gastric cancer. This has lead to bias and high heterogeneity.
- The manuscript should be revised by an English native speaker.
- “helicobacter pylori” should be written out in full the first time it appears (“Helicobacter pylori”). Thereafter, the name should be abbreviated to the initial capital letter (“H. pylori”). In all instances, it should be printed in italics. This should also be considered in the references, if applicable.
- Several paragraphs in the manuscript include lists/numbered sentences. Please revise the manuscript to ensure complete sentences are written.
Introduction
- Unclear why the first and second paragraphs of the introduction focus/provide data for China specifically given that the current manuscript (SR and MA) aims to include and describe studies published worldwide.
- I believe the authors fail to acknowledge the worldwide trends in gastric cancer incidence and mortality. The authors should also update the references to reflect this.
- The authors should mention the findings and recommendations of the World Cancer Research Fund/American Institute for Cancer Research (WCRF/AICR) Report.
- The authors should mention the findings of previous systematic reviews and meta-analyses on salt intake and gastric cancer. Further, what does the current SR and MA add to previously published SR and MA?
- Please also consider the following publication: Salt intake and gastric cancer: a pooled analysis within the Stomach cancer Pooling (StoP) Project
Methods
- “The design, implementation, analysis, and reporting of our meta-analysis were performed in accordance with the PRISMA statement.” Please revise the PRISMA statement provides reporting guidelines.
- How did the authors deal with duplicate publications from the same case-control study? This does not include duplicate records, which should be excluded before screening begins. Which study was selected for inclusion in the current SR and MA?
- Line 79 - Please revise. There is no need to repeat shown in Figure 1. Please revise throughout manuscript.
- Figure 1 should be redone using the PRISMA Flow Diagram (https://prisma-statement.org/PRISMAStatement/FlowDiagram)
- The order in which articles were excluded is incorrect. The same inclusion and exclusion criteria should be applied in steps 1 and 2. Step 1 is screening based on the title and abstract, Step 2 is screening based on the full text. Duplicate records should be excluded before screening begins.
- Line 87 - Please revise “ age deviation”.
Results
- Lines 124-125 - Please specify the relevant tests used to evaluate dietary salt intake. It is unclear what exposures were considered in the studies: salt taste preference, use of table salt, total sodium intake.
- Lines 125-126 - Please specify that the Quality assessment score is provided in Table 1.
- Table 1 - Please review capitalization of the first author name (please do the same for the forest plots), be consistent with writing the last name only, or abbreviating the first name, please remove the extra horizontal lines, please replace age deviation (this metric is not very informative) with mean/median age and standard deviation, interquartile range or range, please consider providing the number of males and females among cases and controls.
- Please provide information on the source of controls (population or hospital based). How may this impact the associations in each study?
- Please provide information on the adjustment variables for each study. A supplementary table may be necessary.
- Please provide the quality assessment (selection of participants, comparability between both groups, and assessment of exposure) of studies in a supplementary document. Consider stratified analyses by quality score (low, high).
- Was less heterogeneity observed when one study was removed at a time? Please discuss.
- Please provide funnel plots in a supplementary document.
- Forest plots - the point estimate circles should be proportional to the weights used in the meta-analysis.
- The authors must explain and describe what they considered in Figure 4 for salt and salty food or salt preference. The authors should not be pooling together estimates that quantify the association between dietary salt exposure defined according to different criteria and gastric cancer, even if the subgroup analysis by estimated methods of salt intake did not show different results.
Discussion
- the authors mention recall bias in Line 74 - this must be further discussed since all studies included are case-control studies. Have the findings from prospective cohort studies been different? Please consider the results from https://doi.org/10.1016/j.clnu.2012.01.003
- the authors state that a limitation of their study is that they cannot clarify the causation, why did the authors opt to only include case-control studies? Did the authors consider also including prospective cohort studies and conducting stratified analyses by study design?
- Consider discussing how the adjustment variables for each study may impact the findings of the SR and MA.
- Lower heterogeneity would have been expected in the stratified analyses. However, there continues to be high heterogeneity. Please discuss.
Conclusions
- It is unclear what this study adds to the current available literature on the association between dietary salt intake and gastric cancer.
- Lines 90-91 - Please review. No cohort studies were included here.
Author Response
Dear professor,
Thank you for your comments concerning our manuscript entitled “Effect of Dietary Salt Intake on Risk of Gastric Cancer: A Systematic Review and Meta-analysis of Case-control Studies” (manuscript ID: nutrients-1889898). Those comments are very valuable and helpful for revising and improving our paper, as well as the important guiding significance to our research. We have carefully revised our manuscript according to those comments, and uploaded our revised manuscript with all the changes highlighted by using the track changes mode. Besides, we provide this cover letter to explain the details of our revisions of the manuscript and our point-by-point responses to the reviewers’ comments. We also have uploaded the modified full text version. We hope that our revised manuscript is acceptable to publication. If you have any questions about our revised manuscript, please inform us without hesitation. Looking forward to hearing from you.
Sincerely,
Jing Wu, MD, PhD
Professor
National Center for Chronic and Non-Communicable Disease Control and Prevention Chinese Center for Disease Control and Prevention
Beijing 100050, China.
E-mail address: [email protected]
Zhongze Fang, MD, PhD
Professor
Department of Toxicology and Sanitary Chemistry
School of Public Health
Tianjin Medical University
Tianjin 300070, China
E-mail address: [email protected]
REVIEWER 3
- The manner in which the systematic review portion of the manuscript was carried out - the authors did not exclude duplicate records correctly, then the authors did not apply exclusion criteria sequentially by screening first the titles and abstracts, and second the full text.
Response: We are grateful for the suggestion. Thanks to you for your good comment. We have redone the process of studies selection. The authors are grateful to the reviewer for pointing out our error. Thank you. We copy it here for your check:
Figure 1. Flow chart of selection of studies for the meta-analysis.
- Studying the association between dietary salt exposure and gastric cancer is subject to several methodological difficulties due to the limitations of the methods available to measure dietary salt exposure. The authors pooled together estimates that quantify the association between dietary salt exposure defined according to different criteria and gastric cancer. This has lead to bias and high heterogeneity. Response: Thank you for your recommendation, we will consider your advice next. Given that the defined criteria of salt exposure are different among each study. We further conduct the subgroup analyses; the results still have high heterogeneity. I thought that the definition is not uniformed, so as the heterogeneity is inevitable. We will continue to conduct relevant study. I hope you will focus on our following study.3. The manuscript should be revised by an English native speaker.Response: Thank you. There are several grammar mistakes in this article. We have corrected these mistakes and revised the manuscript. Thank you.4. “helicobacter pylori” should be written out in full the first time it appears (“Helicobacter pylori”). Thereafter, the name should be abbreviated to the initial capital letter (“H. pylori”). In all instances, it should be printed in italics. This should also be considered in the references, if applicable.Response: Thank you for your suggestion. We have revised relevant part. We copy it here for your check:
The World Cancer Research Fund (WCRF) and its affiliates, including the American Institute for Cancer Research (AICR), have recommended behaviors to prevent cancer, which included healthy diet[5]. Lifestyle factors, including diet, could act over a lifetime to impact cancer risk[5, 6]. High dietary salt intake is one of the leading risk factors for several non-communicable diseases, including gastric cancer[7]. Furthermore, one study suggested that high dietary salt intake is a risk factor for the development of gastric adenocarcinoma[8]. The association may be explained by two important factors. (1) Salt irritates the stomach wall, strongly enhances and promotes chemical gastric carcinogenesis[8, 9]. (2) High salt may increase gastric Helicobacter pylori (Hp) colonization, which is a known risk factor for gastric cancer[10, 11]. High dietary salt intake is also partly responsible for gastric cancer burden worldwide[12, 13]. High Na intake accounted for a large proportion of gastric cancer cases[12]. A health dietary lifestyle is needed. If the optimal lifestyle implemented for all population, half of all gastric cancer events would be prevented by year 2031[6, 14]. If action is taken as early as possible, better effects can be achieved.
Similar to other studies, our study also found that dietary salt intake is associated with gastric cancer. There are several mechanisms to explain this association: (1) salt damages the gastric mucosa directly, leading to hyperplasia of the gastric pit epithelium, and increase the probability of endogenous mutations[52-54]. Additionally, the damage to gastric mucosa could increase DNA damage and glandular atrophy[52].(2) high salt in-take could accelerate the procedure of intestinal metaplasia, which is also an important risk factor of gastric cancer[52]. (3) salty foods often contain too much nitrate and nitrite, which contribute to the formation of N-nitroso compounds[55]. High salt intake may promote or enhance the carcinogenic effect of nitroso compounds[5]. Additionally, high Salt intake may also promote or enhance the effect of other carcinogens[5]. (4) high salt intake increases gastric colonization by H. pylori, one of the main predisposing factors for gastric cancer[5, 52, 53]. One H. pylori virulence determinant that increases gastric cancer risk is the cag pathogenicity island[56]. Cag-positive strains induce more severe gastric injury and further augment the risk for gastric cancer compared with cag-negative strains[56]. Elevated salt concentrations lead to an upregulation of cagA gene in some strains and enhance the ability of cagA to translocate into gastric epithe-lial cells[56, 57]. It indicates that high dietary salt intake could enhance the carcinogenic effects of cagA+ H. pylori strains[52]. (5) High salt intake could alter the viscosity of the protective mucous barrier, disrupt immune homeostasis and increase susceptibility to H. pylori infection[12, 58, 59]. These factors would lead to chronic inflammation, such as atrophic gastritis and gastric ulcer, which are well-accepted precancerous diseases of stomach[52, 53, 58, 60]. 5. Several paragraphs in the manuscript include lists/numbered sentences. Please revise the manuscript to ensure complete sentences are written.Response: Thank you for your valuable advice. We have checked the manuscript and revised related parts. Thank you.
INTRODUCTION
- Unclear why the first and second paragraphs of the introduction focus/provide data for China specifically given that the current manuscript (SR and MA) aims to include and describe studies published worldwide.Response: Thank you for your question. It's really a valuable advice. According to your suggestion, we have re-written and added this part. Please check up again. Thank you. The revised part is as follows.Gastric cancer has been a key public health concern[1]. Gastric cancer incidence and mortality rates showed downward trends past decades, it still is one of the most common cancers and the leading cause of cancer deaths[2, 3]. There were more than 1 million gastric cancer cases in 2020 based on the GLOBOCAN estimates of cancer incidence and mortality, resulting in more than 768,793 deaths[4]. Gastric cancer’s rising prominence as a leading cause of death attracted concern. To prevent or delay the onset of gastric cancer is a prominent strategy.
The World Cancer Research Fund (WCRF) and its affiliates, including the American Institute for Cancer Research (AICR), have recommended behaviors to prevent cancer, which included healthy diet[5]. Lifestyle factors, including diet, could act over a lifetime to impact cancer risk[5, 6]. High dietary salt intake is one of the leading risk factors for several non-communicable diseases, including gastric cancer[7]. Furthermore, one study suggested that high dietary salt intake is a risk factor for the development of gastric adenocarcinoma[8]. The association may be explained by two important factors. (1) Salt irritates the stomach wall, strongly enhances and promotes chemical gastric carcinogenesis[8, 9]. (2) High salt may increase gastric Helicobacter pylori (Hp) colonization, which is a known risk factor for gastric cancer[10, 11]. High dietary salt intake is also partly responsible for gastric cancer burden worldwide[12, 13]. High Na intake accounted for a large proportion of gastric cancer cases[12]. A health dietary lifestyle is needed. If the optimal lifestyle implemented for all population, half of all gastric cancer events would be prevented by year 2031[6, 14]. If action is taken as early as possible, better effects can be achieved.
- I believe the authors fail to acknowledge the worldwide trends in gastric cancer incidence and mortality. The authors should also update the references to reflect this.Response: It's really a valuable advice. According to your suggestion, we have re-written and added this part. Please check up again. Thank you. The revised parts are as follows:Gastric cancer has been a key public health concern[1]. Gastric cancer incidence and mortality rates showed downward trends past decades, it still is one of the most common cancers and the leading cause of cancer deaths[2, 3]. There were more than 1 million gastric cancer cases in 2020 based on the GLOBOCAN estimates of cancer incidence and mortality, resulting in more than 768,793 deaths[4]. Gastric cancer’s rising prominence as a leading cause of death attracted concern. To prevent or delay the onset of gastric cancer is a prominent strategy.
Reference
- Morais S., Costa A., Albuquerque G., Araujo N., Pelucchi C., Rabkin C. S., Liao L. M., Sinha R., Zhang Z. F., Hu J., Johnson K. C., Palli D., Ferraroni M., Bonzi R., Yu G. P., Lopez-Carrillo L., Malekzadeh R., Tsugane S., Hidaka A., Hamada G. S., Zaridze D., Maximovitch D., Vioque J., de la Hera M. G., Moreno V., Vanaclocha-Espi M., Ward M. H., Pakseresht M., Hernandez-Ramirez R. U., Lopez-Cervantes M., Pourfarzi F., Mu L., Kurtz R. C., Boccia S., Pastorino R., Lagiou A., Lagiou P., Boffetta P.,Camargo M. C., Curado M. P., Negri E., La Vecchia. C., Lunet N. Salt intake and gastric cancer: a pooled analysis within the Stomach cancer Pooling (StoP) Project[J]. Cancer Causes Control, 2022,33(5):779-791. Li J., Kuang X H., Zhang Y., Hu D. M., Liu K. Global burden of gastric cancer in adolescents and young adults: estimates from GLOBOCAN 2020[J]. Public Health, 2022,210:58-64.4. Sung H., Ferlay J., Siegel R. L., Laversanne M., Soerjomataram I., Jemal A., Bray F. Global Cancer Statistics 2020: GLOBOCAN Estimates of Incidence and Mortality Worldwide for 36 Cancers in 185 Countries[J]. CA Cancer J Clin, 2021,71(3):209-249. 8. The authors should mention the findings and recommendations of the World Cancer Research Fund/American Institute for Cancer Research (WCRF/AICR) Report.
Response: Thank you for your good advice. According to your suggestion, we have re-written and added this part. Please check up again. Thank you. The revised parts are as follows:
The World Cancer Research Fund (WCRF) and its affiliates, including the American Institute for Cancer Research (AICR), have recommended behaviors to prevent cancer, which included healthy diet[5]. Lifestyle factors, including diet, have acted over a lifetime to impact cancer risk[5, 6]. High dietary salt intake is one of the leading risk factors for several non-communicable diseases, including gastric cancer[7]. Furthermore, one study suggested that high dietary salt intake is a risk factor for the development of gastric adenocarcinoma[8]. The association may be explained by two important factors. (1) Salt irritates the stomach wall, strongly enhances and promotes chemical gastric carcinogenesis[8, 9]. (2) High salt may increase gastric helicobacter pylori (Hp) colonization, which is a known risk factor for gastric cancer[10, 11]. High dietary salt intake is also partly responsible for gastric cancer burden worldwide[12, 13]. High Na intake accounted for a large proportion of gastric cancer cases[12]. A health dietary lifestyle is needed. If the optimal lifestyle implemented for a population, half of all gastric cancer events would be prevented by year 2031[6, 14]. If action is taken as early as possible, better effects can be achieved.
Reference
- Clinton S. K., Giovannucci E. L., Hursting S. D. The World Cancer Research Fund/American Institute for Cancer Research Third Expert Report on Diet, Nutrition, Physical Activity, and Cancer: Impact and Future Directions[J]. J Nutr, 2020,150(4):663-671.
- The authors should mention the findings of previous systematic reviews and meta-analyses on salt intake and gastric cancer. Further, what does the current SR and MA add to previously published SR and MA?
Response: Thank you for the comments. We agree with your proposal. The findings of previous meta-analyses on salt intake and gastric cancer have mentioned in the third paragraphs of the introduction. These meta-analyses were both published in 2012 and the number of included studies was less than current MA. We have supplemented the studies that published between 2012 and 2022 in current MA and further performed subgroup analyses and sensitivity analysis. This MA provide more scientific and theoretical evidence for the further researches. We paste the third paragraphs of the introduction as follows:
Among the previous studies, the association between high dietary salt intake and gastric cancer was established in meta-analysis[15, 16]. But the conclusions of further studies were inconsistent[17-21]. It partly caused by absence of reliable methods for the estimation of dietary salt intake. There are various estimated methods for dietary salt intake, including taste preference, a food frequency questionnaire, dietary behaviors and other methods. The inconsistency of estimated methods may result in the inconsistent results.
Reference
- D'Elia L., Rossi G., Ippolito R., Cappuccio F. P., Strazzullo P. Habitual salt intake and risk of gastric cancer: a meta-analysis of prospective studies[J]. Clin Nutr, 2012,31(4):489-498.
- Ge S., Feng X., Shen L., Wei Z., Zhu Q., Sun J. Association between Habitual Dietary Salt Intake and Risk of Gastric Cancer: A Systematic Review of Observational Studies[J]. Gastroenterol Res Pract, 2012,2012:808120.
- Please also consider the following publication: Salt intake and gastric cancer: a pooled analysis within the Stomach cancer Pooling (StoP) Project.
Response: Thanks to you for your good comment. This is really a meaningful study. Thus, we have added the article as a reference. Please check up again. Thank you.
Reference
- Morais S., Costa A., Albuquerque G., Araujo N., Pelucchi C., Rabkin C. S., Liao L. M., Sinha R., Zhang Z. F., Hu J., Johnson K. C., Palli D., Ferraroni M., Bonzi R., Yu G. P., Lopez-Carrillo L., Malekzadeh R., Tsugane S., Hidaka A., Hamada G. S., Zaridze D., Maximovitch D., Vioque J., de la Hera M. G., Moreno V., Vanaclocha-Espi M., Ward M. H., Pakseresht M., Hernandez-Ramirez R. U., Lopez-Cervantes M., Pourfarzi F., Mu L., Kurtz R. C., Boccia S., Pastorino R., Lagiou A., Lagiou P., Boffetta P.,Camargo M. C., Curado M. P., Negri E., La Vecchia. C., Lunet N. Salt intake and gastric cancer: a pooled analysis within the Stomach cancer Pooling (StoP) Project[J]. Cancer Causes Control, 2022,33(5):779-791.
Thank you for the comments. The introduction refined after revising according your recommendations. The authors are grateful to the reviewer for pointing out our error. Thanks.
METHODS
- “The design, implementation, analysis, and reporting of our meta-analysis were performed in accordance with the PRISMA statement.” Please revise the PRISMA statement provides reporting guidelines.
Response: Thank you for your question. The PRISMA checklist is provided in appendix. We paste it as follows:
Table s1 The PRISMA checklist
Section and Topic |
Item |
Location where item is reported |
TITLE |
||
Title |
1 |
The report is identified as a systematic review and a meta-analysis. |
ABSTRACT |
||
Abstract |
2 |
The structured abstract includes Aim, Methods, Results and Conclusion. |
INTRODUCTION |
||
Rationale |
3 |
Described in the Introduction. |
Objectives |
4 |
Described in the Abstract and the Introduction. |
METHODS |
||
Eligibility criteria |
5 |
They are defined in the Methods. |
Information sources |
6 |
Described in the Methods. |
Search strategy |
7 |
Described in the Methods. |
Selection process |
8 |
Described in the Methods. |
Data collection process |
9 |
Described in the Methods. |
Data items |
10 |
Described in the Methods and summarized in Table 1 and Appendix Table s2. |
Study risk of bias assessment |
11 |
Assessed with Newcastle-Ottawa scale and described in the Methods. Shown in Table 1 and Appendix Table s3. |
Effect measures |
12 |
Odds Ratio. |
Synthesis methods |
13 |
Described in Statistical analysis and reported in detail in Results. |
Reporting bias assessment |
14 |
This have not provided. |
Certainty assessment |
15 |
This have not provided. |
RESULTS |
||
Study selection |
16 |
See Flow Diagram in Figure 1. |
Study characteristics |
17 |
Described in Table 1 and Appendix Table s2. |
Risk of bias in studies |
18 |
Assessed with Newcastle-Ottawa scale and described in Table 1 and Appendix Table s3. |
Results of individual studies |
19 |
Described in Results and shown in Figure 2, Figure 3, Figure 4 and Table 2. |
Results of syntheses |
20 |
Described in Results and shown in Figure 2, Figure 3 and Figure 4. |
Reporting biases |
21 |
Described in the Discussion. |
Certainty of evidence |
22 |
This have not provided. |
DISCUSSION |
||
Discussion |
23 |
All details described in the Discussion. |
OTHER INFORMATION |
||
Registration and protocol |
24 |
The protocol is described in the Methods. The meta-analysis has registered in PROSPERO website (https://www.crd.york.ac.uk/prospero/)(ID: CRD42022354245). |
Support |
25 |
This research was supported by the National Natural Science Foundation of China (grant number: 82273676) and the national key research and development program of China (grant numbers: 2021YFA1301200, 2021YFA1301202). |
Competing interests |
26 |
None. |
Availability of data, code and other materials |
27 |
The data that support the findings of this study are available on request from the corresponding authors. |
- How did the authors deal with duplicate publications from the same case-control study? This does not include duplicate records, which should be excluded before screening begins. Which study was selected for inclusion in the current SR and MA?
Response: Thank you for your advice. There is one point we should clarify. Study with larger sample size was chosen among duplicate publications from the same case-control study. Thank you.
- Line 79 - Please revise. There is no need to repeat shown in Figure 1. Please revise throughout manuscript.Response: Thank you for your advice. We have revised this description throughout manuscript. We copy it here for your check:2.2. Selection criteria and exclusion criteriaThe studies were selected if they met all of the following criteria: (1) being a case-control study; (2) total sample size is over 100; (3) assessment of salty food intake, preference of salty food, use of table salt and relevant indexes as exposure; (4) the au-thors reported the odds ratio (OR) estimates, including 95% confidence intervals (CIs), for different salt intake categories. The studies were excluded if they met any of the following criteria: (1) being systematic reviews, meta-analyses, study protocols, trial registers, meeting abstracts, letters and dissertations without the relevant information; (2) non-human studies. The flow chart of selection of studies was shown in figure 1.3.1. Literature search and study characteristicsFigure 1 shows the study selection process and results from the literature search. Of a total of 1462 publications retrieved (Figure 1), 38 studies were identified that met the inclusion criteria. The relevant characteristics of the 38 studies included in the meta-analysis are reported in Table 1. Overall, the meta-analysis involved 37 225 par-ticipants from 20 countries (11 studies from China, 4 from Korea, 4 from Italy, 2 from Iran, 2 from Turkey, 1 from France, England, Spain, Japan, Puerto Rico, Sweden, Mexico, Thailand, Uruguay, Colombia, Portugal, Serbia, Canada, Ecuador, and Po-land). All the studies recruited both male and female participants. In all studies, the dietary salt intake was estimated by food frequency questionnaires or relevant tests. A summary of characteristics and quality assessment of the included studies is listed in table 1.14. Figure 1 should be redone using the PRISMA Flow Diagram (https://prisma-statement.org/PRISMAStatement/FlowDiagram).
Response: Thanks to you for your good comment. According to your suggestion, we have redone the figure 1 using the PRISMA Flow Diagram. Thank you. We copy it here for your check:
Figure 1. Flow chart of selection of studies for the meta-analysis.
- The order in which articles were excluded is incorrect. The same inclusion and exclusion criteria should be applied in steps 1 and 2. Step 1 is screening based on the title and abstract, Step 2 is screening based on the full text. Duplicate records should be excluded before screening begins.
Response: Thank you for your good advice. We have redone the figure 1. Thank you. We copy it here for your check:
Figure 1. Flow chart of selection of studies for the meta-analysis.
- Line 87 - Please revise “age deviation”.Response: It's really a valuable advice. Thank you. Considering this metric is not very informative, we have replaced age deviation with mean/median age. We copy it here for your check:
Table 1. Characteristics of the case-control studies included in the meta-analysis
First author |
Publication year |
Country |
Region |
Male (n) |
Age (years) Mean/Range |
Sample size |
Quality score |
|
|
|
|
|
|
Case |
Control |
|
|
Tuyns[23] |
1988 |
France |
Europe |
1597 |
— |
— |
4061 |
5 |
Buiatti[24] |
1989 |
Italy |
Europe |
1345 |
≤75 |
≤75 |
2175 |
5 |
Negri[25] |
1990 |
England |
Europe |
219 |
61 |
64 |
282 |
8 |
Demirer[26] |
1990 |
Turkey |
Asia |
131 |
55 |
52 |
200 |
7 |
Hoshiyama[27] |
1992 |
Japan |
Asia |
699 |
— |
— |
699 |
6 |
Ramón[28] |
1993 |
Spain |
Europe |
297 |
62 |
61 |
351 |
8 |
Nazario[29] |
1993 |
Puerto Rico |
America |
— |
≥30 |
≥30 |
271 |
8 |
Hansson[30] |
1993 |
Sweden |
Europe |
662 |
67.7 |
67 |
1017 |
8 |
Lee[31] |
1995 |
Korea |
Asia |
264 |
>25 |
>25 |
426 |
7 |
Vecchia[32] |
1997 |
Italy |
Europe |
1662 |
61 |
55 |
2799 |
5 |
Ye[17] |
1998 |
China |
Asia |
699 |
30-78 |
30-78 |
816 |
8 |
Ji[33] |
1998 |
China |
Asia |
1589 |
61 |
59 |
2575 |
8 |
Ward[34] |
1999 |
Mexico |
America |
— |
≥20 |
≥20 |
972 |
8 |
Palli[35] |
2001 |
Italy |
Europe |
567 |
— |
— |
943 |
6 |
Sriamporn[36] |
2002 |
Thailand |
Asia |
254 |
— |
— |
393 |
7 |
Kim[37] |
2002 |
Korea |
Asia |
186 |
— |
— |
314 |
7 |
Sun[38] |
2002 |
China |
Asia |
568 |
59.8 |
59.5 |
840 |
8 |
Lee[39] |
2003 |
Korea |
Asia |
166 |
— |
— |
268 |
7 |
Stefani[40] |
2004 |
Uruguay |
America |
840 |
30-89 |
30-89 |
1200 |
7 |
Lissowska[41] |
2004 |
Poland |
Europe |
479 |
— |
— |
737 |
8 |
Qiu[42] |
2005 |
China |
Asia |
176 |
63 |
60 |
176 |
6 |
Campos[43] |
2006 |
Colombia |
America |
407 |
— |
— |
647 |
7 |
Hsu[44] |
2008 |
China |
Asia |
131 |
66.0 |
51.8 |
349 |
8 |
Pelucchi[45] |
2009 |
Italy |
Europe |
429 |
63 |
63 |
777 |
7 |
Pourfarzi[46] |
2009 |
Iran |
Asia |
416 |
65.4 |
64.3 |
611 |
8 |
Wen[47] |
2010 |
China |
Asia |
642 |
58.9 |
57.7 |
900 |
7 |
Peleteiro[18] |
2011 |
Portugal |
Europe |
503 |
18-92 |
18-92 |
1071 |
8 |
Yang[48] |
2011 |
China |
Asia |
642 |
52.1 |
52.4 |
900 |
7 |
Lazarević[49] |
2011 |
Serbia |
Europe |
— |
65.8 |
65.8 |
306 |
7 |
Zhang[50] |
2011 |
China |
Asia |
424 |
53.3 |
52.8 |
645 |
6 |
Hu[51] |
2011 |
Canada |
America |
11528 |
57.1 |
60.1 |
6221 |
6 |
Pakseresht[52] |
2011 |
Iran |
Asia |
427 |
66.3 |
62.9 |
590 |
6 |
Yassıbaş[53] |
2012 |
Turkey |
Asia |
132 |
57.4 |
57.9 |
212 |
7 |
Chen[11] |
2012 |
China |
Asia |
390 |
53.1 |
52.8 |
617 |
6 |
Epplein[54] |
2014 |
China |
Asia |
677 |
62.6 |
63.6 |
677 |
8 |
Lin[55] |
2014 |
China |
Asia |
241 |
59.1 |
56.5 |
316 |
6 |
Salvador[19] |
2015 |
Ecuador |
America |
95 |
62.0 |
55.5 |
257 |
7 |
Kwak[21] |
2021 |
Korea |
Asia |
412 |
56.9 |
56.2 |
614 |
7 |
Note: —:not reported or not acquired
RESULTS17. Lines 124-125 - Please specify the relevant tests used to evaluate dietary salt intake. It is unclear what exposures were considered in the studies: salt taste preference, use of table salt, total sodium intake. Response: Thank you. The relevant test is mainly referred to salt taste sensitivity threshold test. All details on exposures were provided in the Table 2. We copy it here for your check:
Table 2 Detailed exposure of the studies
First author |
Publication year |
Match |
The source of controls |
Estimated methods of dietary salt intake |
Comparisons |
Tuyns[23] |
1988 |
non matched |
population based |
addition of salt |
never |
Buiatti[24] |
1989 |
non matched |
population based |
add salt |
never/seldom |
Negri[25] |
1990 |
matched |
population based |
levels of salt intake |
low |
Demirer[26] |
1990 |
matched |
hospital based |
consumption frequency of salted foods |
no consumption/ “rare” consumption/ once or twice a month |
Hoshiyama[27] |
1992 |
non matched |
population based |
preference for salty foods |
low |
Ramón[28] |
1993 |
matched |
population based |
salt intake |
<1.96 (g/d) |
Nazario[29] |
1993 |
non matched |
population based |
salt index |
<6.979(g sodium/w) |
Hansson[30] |
1993 |
matched |
population based |
salted fish |
low |
Lee[31] |
1995 |
non matched |
hospital based |
salt preference |
low |
Vecchia[32] |
1997 |
non matched |
hospital based |
salt preference |
low |
Ye[17] |
1998 |
matched |
population based |
salt |
≤0.25kg/m |
Ji[33] |
1998 |
matched |
population based |
Consume salted foods |
occasionally |
Ward[34] |
1999 |
non matched |
population based |
Salty snacks/crackers |
never |
Palli[35] |
2001 |
non matched |
population based |
sodium intake |
low tertile |
Sriamporn[36] |
2002 |
matched |
hospital based |
salted food |
low |
Kim[37] |
2002 |
matched |
hospital based |
salted food |
low |
Sun[38] |
2002 |
matched |
population based |
salt preference |
moderate |
Lee[39] |
2003 |
non matched |
hospital based |
salt fermented fish |
<1/month |
Stefani[40] |
2004 |
matched |
hospital based |
salted meat consumption |
low |
Lissowska[41] |
2004 |
matched |
population based |
weekly frequency of salt consumption |
low |
Qiu[42] |
2005 |
non matched |
population based |
daily intake of sodium |
low |
Campos[43] |
2006 |
matched |
hospital based |
salting meals before tasting |
no |
Hsu[44] |
2008 |
non matched |
population based |
salty food intake |
low |
Pelucchi[45] |
2009 |
matched |
hospital based |
intake of sodium |
low |
Pourfarzi[46] |
2009 |
matched |
population based |
salt preference |
not salty |
Wen[47] |
2010 |
matched |
hospital based |
STST≥5 |
STST<5 |
Peleteiro[18] |
2011 |
non matched |
population based |
use of table salt (salt consumption by visual analogical scale) |
<35(mm) |
Yang[48] |
2011 |
matched |
hospital based |
salt taste preference |
not salty |
Lazarević[49] |
2011 |
matched |
hospital based |
intake of salt |
low |
Zhang[50] |
2011 |
non matched |
population based |
salt taste preference* |
0.9 (g/L) |
Hu[51] |
2011 |
non matched |
population based |
added salt at table |
never |
Pakseresht[52] |
2011 |
non matched |
population based |
salt |
per g |
Yassıbaş[53] |
2012 |
matched |
hospital based |
salt status of dishes |
Salt free |
Chen[11] |
2012 |
non matched |
population based |
salt taste preference* |
<1.8 |
Epplein[54] |
2014 |
matched |
population based |
intake of sodium |
low |
Lin[55] |
2014 |
matched |
hospital based |
salt taste preference |
not salty |
Salvador[19] |
2015 |
non matched |
hospital based |
adding salt >50% of meals |
no |
Kwak[21] |
2021 |
matched |
hospital based |
salt taste preference |
no opinion |
Note: STST: Salt taste sensitivity threshold, d:day, w:week, *: the salt preference was assessed by threshold level of salty taste
- Lines 125-126 - Please specify that the Quality assessment score is provided in Table 1.
Response: Thank you for your question. The Newcastle-Ottawa Scale was used in this meta-analysis. The detailed description has supplemented in the text. The process of quality assessment has provided in the supplementary document. We copy it here for your check:
Figure 1 shows the study selection process and results from the literature search. Of a total of 1462 publications retrieved (Figure 1), 38 studies were identified that met the inclusion criteria. The relevant characteristics of the 38 studies included in the meta-analysis are reported in Table 1. Overall, the meta-analysis involved 37 225 par-ticipants from 20 countries (11 studies from China, 4 from Korea, 4 from Italy, 2 from Iran, 2 from Turkey, 1 from France, England, Spain, Japan, Puerto Rico, Sweden, Mexico, Thailand, Uruguay, Colombia, Portugal, Serbia, Canada, Ecuador, and Po-land). All the studies recruited both male and female participants. In all studies, the dietary salt intake was estimated by food frequency questionnaires or relevant tests. The Newcastle-Ottawa Scale was performed to assess the quality of included articles. A summary of characteristics and quality assessment of the included studies is listed in table 1 (shown in table 1). 19. Table 1 - Please review capitalization of the first author name (please do the same for the forest plots), be consistent with writing the last name only, or abbreviating the first name, please remove the extra horizontal lines, please replace age deviation (this metric is not very informative) with mean/median age and standard deviation, interquartile range or range, please consider providing the number of males and females among cases and controls.
Response: Thank you for your good advice. We have redone the Table 1. Thank you. We copy it here for your check:
Table 1. Characteristics of the case-control studies included in the meta-analysis
First author |
Publication year |
Country |
Region |
Male (n) |
Age (years) Mean/Range |
Sample size |
Quality score |
|
|
|
|
|
|
Case |
Control |
|
|
Tuyns[23] |
1988 |
France |
Europe |
1597 |
— |
— |
4061 |
5 |
Buiatti[24] |
1989 |
Italy |
Europe |
1345 |
≤75 |
≤75 |
2175 |
5 |
Negri[25] |
1990 |
England |
Europe |
219 |
61 |
64 |
282 |
8 |
Demirer[26] |
1990 |
Turkey |
Asia |
131 |
55 |
52 |
200 |
7 |
Hoshiyama[27] |
1992 |
Japan |
Asia |
699 |
— |
— |
699 |
6 |
Ramón[28] |
1993 |
Spain |
Europe |
297 |
62 |
61 |
351 |
8 |
Nazario[29] |
1993 |
Puerto Rico |
America |
— |
≥30 |
≥30 |
271 |
8 |
Hansson[30] |
1993 |
Sweden |
Europe |
662 |
67.7 |
67 |
1017 |
8 |
Lee[31] |
1995 |
Korea |
Asia |
264 |
>25 |
>25 |
426 |
7 |
Vecchia[32] |
1997 |
Italy |
Europe |
1662 |
61 |
55 |
2799 |
5 |
Ye[17] |
1998 |
China |
Asia |
699 |
30-78 |
30-78 |
816 |
8 |
Ji[33] |
1998 |
China |
Asia |
1589 |
61 |
59 |
2575 |
8 |
Ward[34] |
1999 |
Mexico |
America |
— |
≥20 |
≥20 |
972 |
8 |
Palli[35] |
2001 |
Italy |
Europe |
567 |
— |
— |
943 |
6 |
Sriamporn[36] |
2002 |
Thailand |
Asia |
254 |
— |
— |
393 |
7 |
Kim[37] |
2002 |
Korea |
Asia |
186 |
— |
— |
314 |
7 |
Sun[38] |
2002 |
China |
Asia |
568 |
59.8 |
59.5 |
840 |
8 |
Lee[39] |
2003 |
Korea |
Asia |
166 |
— |
— |
268 |
7 |
Stefani[40] |
2004 |
Uruguay |
America |
840 |
30-89 |
30-89 |
1200 |
7 |
Lissowska[41] |
2004 |
Poland |
Europe |
479 |
— |
— |
737 |
8 |
Qiu[42] |
2005 |
China |
Asia |
176 |
63 |
60 |
176 |
6 |
Campos[43] |
2006 |
Colombia |
America |
407 |
— |
— |
647 |
7 |
Hsu[44] |
2008 |
China |
Asia |
131 |
66.0 |
51.8 |
349 |
8 |
Pelucchi[45] |
2009 |
Italy |
Europe |
429 |
63 |
63 |
777 |
7 |
Pourfarzi[46] |
2009 |
Iran |
Asia |
416 |
65.4 |
64.3 |
611 |
8 |
Wen[47] |
2010 |
China |
Asia |
642 |
58.9 |
57.7 |
900 |
7 |
Peleteiro[18] |
2011 |
Portugal |
Europe |
503 |
18-92 |
18-92 |
1071 |
8 |
Yang[48] |
2011 |
China |
Asia |
642 |
52.1 |
52.4 |
900 |
7 |
Lazarević[49] |
2011 |
Serbia |
Europe |
— |
65.8 |
65.8 |
306 |
7 |
Zhang[50] |
2011 |
China |
Asia |
424 |
53.3 |
52.8 |
645 |
6 |
Hu[51] |
2011 |
Canada |
America |
11528 |
57.1 |
60.1 |
6221 |
6 |
Pakseresht[52] |
2011 |
Iran |
Asia |
427 |
66.3 |
62.9 |
590 |
6 |
Yassıbaş[53] |
2012 |
Turkey |
Asia |
132 |
57.4 |
57.9 |
212 |
7 |
Chen[11] |
2012 |
China |
Asia |
390 |
53.1 |
52.8 |
617 |
6 |
Epplein[54] |
2014 |
China |
Asia |
677 |
62.6 |
63.6 |
677 |
8 |
Lin[55] |
2014 |
China |
Asia |
241 |
59.1 |
56.5 |
316 |
6 |
Salvador[19] |
2015 |
Ecuador |
America |
95 |
62.0 |
55.5 |
257 |
7 |
Kwak[21] |
2021 |
Korea |
Asia |
412 |
56.9 |
56.2 |
614 |
7 |
Note: —:not reported or not acquired
- Please provide information on the source of controls (population or hospital based). How may this impact the associations in each study?Response: Thank you for your comment. The information on the source of controls have provided in Table 2. Subgroup analyses were further used to identify associations. stratifying by the source of controls estimated methods, the pooled ORs were 1.39 (95% CI, [1.30, 1.49]) for studies conducted in population-based controls, and 2.19 (95% CI, [1.93, 2.49]) for studies conducted in hospital-based controls. The detailed information has provided in Figure 5. We copy it here for your check:
Table 2 Detailed exposure of the studies
First author |
Publication year |
Match |
The source of controls |
Estimated methods of dietary salt intake |
Comparisons |
Tuyns[23] |
1988 |
non matched |
population based |
addition of salt |
never |
Buiatti[24] |
1989 |
non matched |
population based |
add salt |
never/seldom |
Negri[25] |
1990 |
matched |
population based |
levels of salt intake |
low |
Demirer[26] |
1990 |
matched |
hospital based |
consumption frequency of salted foods |
no consumption/ “rare” consumption/ once or twice a month |
Hoshiyama[27] |
1992 |
non matched |
population based |
preference for salty foods |
low |
Ramón[28] |
1993 |
matched |
population based |
salt intake |
<1.96 (g/d) |
Nazario[29] |
1993 |
non matched |
population based |
salt index |
<6.979(g sodium/w) |
Hansson[30] |
1993 |
matched |
population based |
salted fish |
low |
Lee[31] |
1995 |
non matched |
hospital based |
salt preference |
low |
Vecchia[32] |
1997 |
non matched |
hospital based |
salt preference |
low |
Ye[17] |
1998 |
matched |
population based |
salt |
≤0.25kg/m |
Ji[33] |
1998 |
matched |
population based |
Consume salted foods |
occasionally |
Ward[34] |
1999 |
non matched |
population based |
Salty snacks/crackers |
never |
Palli[35] |
2001 |
non matched |
population based |
sodium intake |
low tertile |
Sriamporn[36] |
2002 |
matched |
hospital based |
salted food |
low |
Kim[37] |
2002 |
matched |
hospital based |
salted food |
low |
Sun[38] |
2002 |
matched |
population based |
salt preference |
moderate |
Lee[39] |
2003 |
non matched |
hospital based |
salt fermented fish |
<1/month |
Stefani[40] |
2004 |
matched |
hospital based |
salted meat consumption |
low |
Lissowska[41] |
2004 |
matched |
population based |
weekly frequency of salt consumption |
low |
Qiu[42] |
2005 |
non matched |
population based |
daily intake of sodium |
low |
Campos[43] |
2006 |
matched |
hospital based |
salting meals before tasting |
no |
Hsu[44] |
2008 |
non matched |
population based |
salty food intake |
low |
Pelucchi[45] |
2009 |
matched |
hospital based |
intake of sodium |
low |
Pourfarzi[46] |
2009 |
matched |
population based |
salt preference |
not salty |
Wen[47] |
2010 |
matched |
hospital based |
STST≥5 |
STST<5 |
Peleteiro[18] |
2011 |
non matched |
population based |
use of table salt (salt consumption by visual analogical scale) |
<35(mm) |
Yang[48] |
2011 |
matched |
hospital based |
salt taste preference |
not salty |
Lazarević[49] |
2011 |
matched |
hospital based |
intake of salt |
low |
Zhang[50] |
2011 |
non matched |
population based |
salt taste preference* |
0.9 (g/L) |
Hu[51] |
2011 |
non matched |
population based |
added salt at table |
never |
Pakseresht[52] |
2011 |
non matched |
population based |
salt |
per g |
Yassıbaş[53] |
2012 |
matched |
hospital based |
salt status of dishes |
Salt free |
Chen[11] |
2012 |
non matched |
population based |
salt taste preference* |
<1.8 |
Epplein[54] |
2014 |
matched |
population based |
intake of sodium |
low |
Lin[55] |
2014 |
matched |
hospital based |
salt taste preference |
not salty |
Salvador[19] |
2015 |
non matched |
hospital based |
adding salt >50% of meals |
no |
Kwak[21] |
2021 |
matched |
hospital based |
salt taste preference |
no opinion |
Note: STST: Salt taste sensitivity threshold, d:day, w:week, *: the salt preference was assessed by threshold level of salty taste
Figure 5 Forest plot of associations between high dietary salt intake and gastric cancer risk among the source of controls.
- Please provide information on the adjustment variables for each study. A supplementary table may be necessary.Response: Thank you for your question. The information on the adjustment variables have provided in a supplementary document. We copy it here for your check:
Table s2. Adjustment variables of the case-control studies included in the meta-analysis.
First author |
Publication year |
Adjustment variables |
Tuyns[19] |
1988 |
age, sex, and province |
Buiatti[20] |
1989 |
age, sex, area, place of residence, migration from the south, socio-economic status, familial history of GC, Quetelet index, tertile levels of consumption of one or more dietary variables |
Negri[21] |
1990 |
not refer |
Demirer[22] |
1990 |
not refer |
Hoshiyama[23] |
1992 |
not refer |
Ramón[24] |
1993 |
sex, age, education, cigarettes/day, rice, citrus fruit, raw green vegetables, all fruits, cereals, smoked and pickled foods |
Nazario[25] |
1993 |
not refer |
Hansson[26] |
1993 |
age, gender, socio-economic status |
Lee[27] |
1995 |
age, sex, education, economic status, residence, other dietary factors |
Vecchia[28] |
1997 |
sex, age, education |
Ye[13] |
1998 |
not refer |
Ji[29] |
1998 |
age, sex, income, education, smoking, alcohol drinking |
Ward[30] |
1999 |
age, gender, total calories, chili pepper consumption, added salt, history of peptic ulcer, cigarette smoking, and socioeconomic status |
Palli[31] |
2001 |
age, sex, social class, family history of gastric cancer, area of rural residence, BMI tertiles, total energy, tertiles of the residuals of each nutrient of interest. |
Sriamporn[32] |
2002 |
age, sex, fermentated food |
Kim[33] |
2002 |
sex, age, socioeconomic status, family history and refrigerator use |
Sun[34] |
2002 |
age, income, resident space, using refrigerator and educational level |
Lee[35] |
2003 |
age, sex, education, family history of GC, smoking, drinking, H. pylori infection |
Stefani[36] |
2004 |
Age, sex, residence, urban/rural status, education, BMI, total energy intake |
Lissowska[37] |
2004 |
age, sex, education, smoking, calories from foods |
Qiu[38] |
2005 |
age, present residence, education, economic status, smoking, alcoholics, total calories intake |
Campos[39] |
2006 |
not refer |
Hsu[40] |
2008 |
carriage of myeloperoxidase allele A, gender, advanced age, tea consumption, level of education, H. pylori infection |
Pelucchi[41] |
2009 |
Age, sex, adjusted for period of interview, education, BMI, tobacco smoking, family history of stomach cancer, total energy intake |
Poufarzi[42] |
2009 |
gender, age group, education, family history of GC, citrus fruits, garlic, onion, red, meat, fish, dairy products, strength and warmth of tea, preference for salt intake and H. pylori |
Wen[43] |
2010 |
age, sex, BMI, family history, smoking, drinking, fresh fruit, fresh vegetables |
Peleteiro[14] |
2011 |
gender, age, education, smoking and H. pylori infection |
Yang[44] |
2011 |
age, sex, smoking, drinking, fresh fruit and fresh vegetables; |
Lazarević[45] |
2011 |
not refer |
Zhang[46] |
2011 |
sex, age, education level, smoking, drinking, H. pylori infection |
Hu[47] |
2011 |
age group, province, education, BMI, sex, alcohol drinking, pack-years smoking, total vegetable and fruit intake, total energy intake; |
Pakseresht[48] |
2011 |
age, sex, education, living area, smoking, gastric symptoms, income, owning refrigerator, duration of using refrigerator, seeds preparing method, frying, H. pylori infection, total energy intake |
Yassıbaş[49] |
2012 |
gender, residence, education, smoking, alcohol consumption and family history of cancer for 26 kinds of foods considered to be related to gastric cancer |
Zhong[10] |
2012 |
sex, age, education level, smoking, drinking, H. pylori infection |
Epplein[50] |
2014 |
age, smoking, history of gastritis, regular aspirin use, average, total energy intake |
Lin[51] |
2014 |
age, sex, home income, family history of cancer, smoking status, alcohol drinking, fresh vegetables intake, fresh fruit intake |
Salvador[15] |
2015 |
not refer |
Kwak[17] |
2021 |
Age, sex, BMI, education level, family history of gastric cancer, smoking status, alcohol drinkers, total energy intake, H. pylori infection |
Abbreviations: GC, gastric cancer, H. pylori: Helicobacter pylori, BMI: body mass index.
- Please provide the quality assessment (selection of participants, comparability between both groups, and assessment of exposure) of studies in a supplementary document. Consider stratified analyses by quality score (low, high).Response: Thank you for your question. The information on the quality assessment have provided in a supplementary document. The score is determined by researchers. Everyone's opinion is bound to be subjective. The explanation above is the reason why we did not provide the stratified analyses by quality score. Thank you. We copy it here for your check:
Table S3 The study quality scores of the studies included in meta-analysis
|
Selection |
|
Comparability |
|
Exposure |
|||||
First author |
Is the case adequate definition |
Representativeness of the cases |
Selection of Controls |
Definition of Controls |
|
Comparability of cases and controls on the basis of the design or analysis |
|
Ascertainment of exposure |
Same method of ascertainment for cases and controls |
Non-Response rate |
Tuyns[21] |
0 |
* |
* |
* |
|
0 |
|
* |
* |
0 |
Buiatti[22] |
* |
* |
* |
* |
|
0 |
|
* |
* |
0 |
Negri[23] |
* |
* |
* |
* |
|
** |
|
* |
* |
0 |
Demirer[24] |
* |
* |
0 |
* |
|
** |
|
* |
* |
0 |
Hoshiyama[25] |
* |
* |
* |
* |
|
0 |
|
* |
* |
0 |
Ramón[26] |
* |
* |
* |
* |
|
** |
|
* |
* |
0 |
Nazario[27] |
* |
* |
* |
* |
|
** |
|
* |
* |
0 |
Hansson[28] |
* |
* |
* |
* |
|
** |
|
* |
* |
0 |
Lee[29] |
* |
* |
0 |
* |
|
** |
|
* |
* |
0 |
Vecchia[30] |
* |
* |
0 |
* |
|
0 |
|
* |
* |
0 |
Ye[15] |
* |
* |
* |
* |
|
** |
|
* |
* |
0 |
Ji[31] |
* |
* |
* |
* |
|
** |
|
* |
* |
0 |
Ward[32] |
* |
* |
* |
* |
|
* |
|
* |
* |
* |
Palli[33] |
* |
* |
* |
* |
|
0 |
|
* |
* |
0 |
Sriamporn[34] |
* |
* |
0 |
* |
|
** |
|
* |
* |
0 |
Kim[35] |
* |
* |
0 |
* |
|
** |
|
* |
* |
0 |
Sun[36] |
* |
* |
* |
* |
|
** |
|
* |
* |
0 |
Lee[37] |
* |
* |
0 |
* |
|
** |
|
* |
* |
0 |
Stefani[38] |
* |
* |
0 |
* |
|
** |
|
* |
* |
0 |
Lissowska[39] |
* |
* |
* |
* |
|
** |
|
* |
* |
0 |
Qiu[40] |
* |
* |
* |
* |
|
0 |
|
* |
* |
0 |
Campos[40] |
* |
* |
0 |
* |
|
** |
|
* |
* |
0 |
Hsu[42] |
* |
* |
* |
* |
|
** |
|
* |
* |
0 |
Pelucchi[43] |
* |
* |
0 |
* |
|
** |
|
* |
* |
0 |
Pourfarzi[44] |
* |
* |
* |
* |
|
** |
|
* |
* |
0 |
Wen[45] |
* |
* |
0 |
* |
|
** |
|
* |
* |
0 |
Peleteiro[16] |
* |
* |
* |
* |
|
** |
|
* |
* |
0 |
Yang[46] |
* |
* |
0 |
* |
|
** |
|
* |
* |
0 |
Lazarević[47] |
* |
* |
0 |
* |
|
** |
|
* |
* |
0 |
Zhang[48] |
* |
* |
* |
* |
|
0 |
|
* |
* |
0 |
Hu[49] |
* |
* |
* |
* |
|
0 |
|
* |
* |
0 |
Pakseresht[50] |
* |
* |
* |
* |
|
0 |
|
* |
* |
0 |
Yassıbaş[51] |
* |
* |
0 |
* |
|
** |
|
* |
* |
0 |
Chen [11] |
* |
* |
* |
* |
|
0 |
|
* |
* |
0 |
Epplein[52] |
* |
* |
* |
* |
|
** |
|
* |
* |
0 |
Lin[53] |
* |
* |
0 |
* |
|
** |
|
* |
0 |
0 |
Salvador[17] |
* |
* |
0 |
* |
|
** |
|
* |
* |
0 |
Kwak[19] |
* |
* |
0 |
* |
|
** |
|
* |
* |
0 |
23. Was less heterogeneity observed when one study was removed at a time? Please discuss.Response: Thank you for your comment. The change of heterogeneity was explored when one study was removed at a time. The results indicated that variation fluctuated between 82.1% and 83.3%. We failed to explore effect of single study on heterogeneity. 24. Please provide funnel plots in a supplementary document.Response: Thank you for your question. The funnel plots have supplemented in a supplementary document. We copy it here for your check: Figure s1. Funnel plots for identifying publication bias in the meta-analysis of observational studies. Abbreviations: OR, odds ratio. 25. Forest plots - the point estimate circles should be proportional to the weights used in the meta-analysis.
Response: Thank you for your good advice. We have redone the figure 2. Thank you. We copy it here for your check:
Figure 2 Forest plot of associations between high dietary salt intake and gastric cancer risk.
26. The authors must explain and describe what they considered in Figure 4 for salt and salty food or salt preference. The authors should not be pooling together estimates that quantify the association between dietary salt exposure defined according to different criteria and gastric cancer, even if the subgroup analysis by estimated methods of salt intake did not show different results.Response: Thank you for your question. I must clarify this point. In our paper, we explained and described the findings of figure 4 in discussion. we will consider your advice next, pooling together estimates that quantify the association between dietary salt exposure defined according to different criteria and gastric cancer. I hope you will focus on our following study. DISCUSSION28. the authors mention recall bias in Line 74 - this must be further discussed since all studies included are case-control studies. Have the findings from prospective cohort studies been different? Please consider the results from https://doi.org/10.1016/j.clnu.2012.01.003
Response: Thank you for your good advice. The meta-analysis that referred above also founded a small but significant association between salt intake and gastric cancer. Estimated methods of dietary salt intake in this meta-analysis focused on salty foods. According to your suggestion, we have re-written and added this part. Please check up again. Thank you. The revised parts are as follows:
There was a significant heterogeneity among the included studies. This situation also existed in other similar studies[15, 16]. Further subgroup analyses were carried out to check for potential sources of heterogeneity, which might explain the association between dietary salt intake and gastric cancer events. Among the studies that estimated salt intake by consumption of salty foods or salt preference, the heterogeneity has been decreased. It indicated that estimated methods for dietary salt intake maybe a source of heterogeneity. It is difficult to quantify the intake of sodium, main ingredient of salt. The food frequency questionnaire (FFQ) was used to estimate the dietary salt intake in most studies. The actual intake amount of salt could not estimate through FFQ, and the recall bias is inevitable. Cases tend to overestimate their exposure to risk factors possibly, this may lead to a spurious association between risk factors and disease[63].
29. the authors state that a limitation of their study is that they cannot clarify the causation, why did the authors opt to only include case-control studies? Did the authors consider also including prospective cohort studies and conducting stratified analyses by study design?
Response: Thank you for your recommendation, we will consider your advice next. The findings of previous meta-analyses on salt intake and gastric cancer were both published in 2012 and the number of included studies was less than current MA. We have supplemented the studies that published between 2012 and 2022 in current MA and further performed subgroup analyses and sensitivity analysis. This MA provide more scientific and theoretical evidence for the further researches. We have considered including prospective cohort studies. I hope you will focus on our following study.
30. Consider discussing how the adjustment variables for each study may impact the findings of the SR and MA.
Response: Thank you for your recommendation, we will consider your advice next. The information on the adjustment variables have provided in a supplementary document. I hope you will focus on our following study. Thank you.
31. Lower heterogeneity would have been expected in the stratified analyses. However, there continues to be high heterogeneity. Please discuss.Response: Thank you for your question. I must clarify this point. We explained and described the high heterogeneity in discussion. Thank you. CONCLUSION32. It is unclear what this study adds to the current available literature on the association between dietary salt intake and gastric cancer.
Response: Thank you for the comments. The previous meta-analyses were both published in 2012 and the number of included studies was less than current MA. We have supplemented the studies that published between 2012 and 2022 in current MA and further performed subgroup analyses and sensitivity analysis. This MA provide more scientific and theoretical evidence for the further researches. Thank you.
33. Lines 90-91 - Please review. No cohort studies were included here.Response: Thank you for your question. I must clarify this point. This Meta-analysis have included 38 case-control studies. Similar to numerous reports in the last 10 years, we founded a small but significant association between dietary sodium intake and gastric cancer. However, this association is not clear. We thought more meta-analysis included cohort studies is responsible for cause-effect clarification. Thank you. We tried our best to improve the manuscript and made some changes in the manuscript. These changes will not influence the content and framework of the paper. We appreciate the Editors/Reviewers’ warm work earnestly, and hope that the correction will meet with approval. Once again, thank you very much for your comments and suggestions.

Round 2
Reviewer 3 Report
I appreciate the authors efforts in improving the manuscript. I still have a few concerns. Please see below.
- The manuscript needs to be revised by a native English speaker.
- “2.2. Selection criteria and exclusion criteria” - the authors must state and clarify that they applied exclusion criteria sequentially by screening first the titles and abstracts, and second the full text. Further, the exclusion criteria listed in lines 79-82 do not match those presented in Figure 1.
- Line 45 - please remove “(Hp)”. The name should be abbreviated to the initial capital letter (“H. pylori”) and printed in italics.
- Line 89 “H. pylori” should be printed in italics.
- Several paragraphs in the manuscript continue to include lists/numbered sentences, in which the first word of a sentence is not capitalized. Please revise the manuscript again to ensure complete sentences are written.
- Lines 30 - 35 - References 1 and 2 are not appropriate.
- The authors added the recent publication from the StoP Project, however, the reference is not used in a relevant manner. Please exclude or revise.
- Lines 63-64 please revise - the PRISMA statement provides reporting guidelines not “performing” guidelines.
- Regarding: “12. How did the authors deal with duplicate publications from the same case-control study? This does not include duplicate records, which should be excluded before screening begins. Which study was selected for inclusion in the current SR and MA?
Response: Thank you for your advice. There is one point we should clarify. Study with larger sample size was chosen among duplicate publications from the same case-control study. Thank you.” — No changes have been made to the manuscript. This information must be provided to readers.
- Regarding “15. The order in which articles were excluded is incorrect. The same inclusion and exclusion criteria should be applied in steps 1 and 2. Step 1 is screening based on the title and abstract, Step 2 is screening based on the full text. Duplicate records should be excluded before screening begins.” - The authors made changes to figure 1. Flow chart, however no changes have been made to the manuscript text.
- Regarding: “17. Lines 124-125 - Please specify the relevant tests used to evaluate dietary salt intake. It is unclear what exposures were considered in the studies: salt taste preference, use of table salt, total sodium intake. “ - the authors have added table 2 which provides information for this, however, no changes gave been made to the manuscript text.
- Regarding: “18. Lines 125-126 - Please specify that the Quality assessment score is provided in Table 1.” - the authors provide the quality assessment in a supplementary table and the authors changed the manuscript text: “The Newcastle-Ottawa Scale was performed to assess the quality of included articles. ”, however, they do not state that the reader should refer to the supplementary table for these results/information. Please revise accordingly.
- Regarding: “20. Please provide information on the source of controls (population or hospital based). How may this impact the associations in each study?” - Please discuss the difference in the estimates (1.39 for population based controls, and 2.19 for hospital based controls). Figure 5 is not included in the manuscript (could be a supplementary figure) and results for the different controls are not provided.
- Regarding: “21. Please provide information on the adjustment variables for each study. A supplementary table may be necessary.” - the authors have added Table S2, however this is not ever mentioned in the manuscript text. Readers will not know this information is available unless it’s mentioned in the manuscript text.
- Regarding: “30. Consider discussing how the adjustment variables for each study may impact the findings of the SR and MA.” - the authors have included a supplementary table with information on adjustment variables, however they fail to mention it in the manuscript (the reader will not know it is available), and fail to discuss how adjustment variables for each study may impact their results - if studies used many different adjustment variables - this may have potentially led to increased heterogeneity. I appreciate the authors highlighting that certain risk factors were not considered as adjustment variables (Lines 89-90).
Author Response
Dear professor,
Thank you for your comments concerning our manuscript entitled “Effect of Dietary Salt Intake on Risk of Gastric Cancer: A Systematic Review and Meta-analysis of Case-control Studies” (manuscript ID: nutrients-1889898). Those comments are very valuable and helpful for revising and improving our paper, as well as the important guiding significance to our research. We have carefully revised our manuscript according to those comments, and uploaded our revised manuscript with all the changes highlighted by using the track changes mode. Besides, we provide this cover letter to explain the details of our revisions of the manuscript and our point-by-point responses to the reviewers’ comments. We also have uploaded the modified full text version. We hope that our revised manuscript is acceptable to publication. If you have any questions about our revised manuscript, please inform us without hesitation. Looking forward to hearing from you.
Sincerely,
Jing Wu, MD, PhD
Professor
National Center for Chronic and Non-Communicable Disease Control and Prevention Chinese Center for Disease Control and Prevention
Beijing 100050, China.
E-mail address: [email protected]
Zhongze Fang, MD, PhD
Professor
Department of Toxicology and Sanitary Chemistry
School of Public Health
Tianjin Medical University
Tianjin 300070, China
E-mail address: [email protected]
- The manuscript needs to be revised by a native English speaker.
Response: Thank you. We have improved the English in the secondly revised manuscript. Thank you.
- “2.2. Selection criteria and exclusion criteria” - the authors must state and clarify that they applied exclusion criteria sequentially by screening first the titles and abstracts, and second the full text. Further, the exclusion criteria listed in lines 79-82 do not match those presented in Figure 1.
Response: Thanks to you for your good comment. According to your suggestion, we have re-written the related part. Thank you. We copy it here for your check:
2.2. Selection criteria and exclusion criteria
The studies were selected if they met all of the following criteria: (1) being a case-control study; (2) total sample size is over 100; (3) assessment of salty food intake, preference of salty food, use of table salt and relevant indexes as exposure; (4) the authors reported the odds ratio (OR) estimates, including 95% confidence intervals (CIs), for different salt intake categories. The studies were excluded if they met any of the following criteria: (1) being duplicate publications; (2) not relevant; (3) being systematic reviews, meta-analyses, meeting abstracts, letters, and dissertations without the relevant information; (4) not case-control studies; (5) OR and 95%CI not be reported. Study with larger sample size was chosen among duplicate publications from the same case-control study. Exclusion criteria were applied sequentially by screening first the titles and abstracts, and second the full text. Duplicate records were excluded before screening begins. The flow chart of the selection of studies was shown in figure 1.
- Line 45 - please remove “(Hp)”. The name should be abbreviated to the initial capital letter (“H. pylori”) and printed in italics.
Response: Thank you for your advice. We have revised this description throughout manuscript. We copy it here for your check:
The World Cancer Research Fund (WCRF) and its affiliates, including the American Institute for Cancer Research (AICR), have suggested cancer-prevention behaviors such as a healthy diet [3]. Lifestyle factors, including diet, may have an impact on cancer risk over a lifetime [3,4]. High salt consumption is one of the leading risk factors for a variety of non-communicable diseases, including gastric cancer [5]. Furthermore, one study founded that a high salt intake may be a risk factor for the development of gastric adenocarcinoma [6]. The association may be explained by two important factors. (1) Salt irritates the stomach wall and strongly enhances and promotes chemical gastric carcinogenesis [6,7]. (2) Excess salt may promote gastric Helicobacter pylori (H. pylori) colonization in the stomach, which is a known risk factor for gastric cancer [8,9]. High dietary salt intake is also contributing to the global burden of gastric cancer [10,11]. High sodium intake accounted for much of gastric cancer cases [11]. A healthy diet and lifestyle are required. By implementing the optimal lifestyle for all populations, half of all gastric cancer events could be prevented by the year 2031 [3,12]. If action is taken as early as possible, better effects can be achieved.
- Line 89 “H. pylori” should be printed in italics.
Response: Thank you for your advice. We have revised this description throughout manuscript. We copy it here for your check:
There exist several potential limitations in this study. First, we only included studies published in English. And we did not search grey literatures. The actual total number of included studies maybe larger than the current included studies. Second, confounding risk factors such as H. pylori, smoking, and other relevant risk factors were not able to be considered in this meta-analysis. Third, given the observational nature of the included studies, our study lacked evidence to clarify the causation. Fourth, the estimated methods of dietary salt intake, which contributed to the heterogeneity of this study, were not classified in more detail.
- Several paragraphs in the manuscript continue to include lists/numbered sentences, in which the first word of a sentence is not capitalized. Please revise the manuscript again to ensure complete sentences are written.
Response: Thank you for your advice. We have revised this description throughout manuscript. We copy it here for your check:
Similar to other studies, our study also found that dietary salt intake is associated with gastric cancer. There are several mechanisms to explain this association: (1) The gastric mucosa could be damaged by high salt concentration directly, which leads to hyperplasia of the gastric pit epithelium and increase the probability of endogenous mutations [13,14,53]. Additionally, the damage to gastric mucosa could increase DNA damage and glandular atrophy [14]. (2) High salt intake could accelerate the procedure of intestinal metaplasia, which could develop into early gastric cancer [14]. (3) Salty foods that are of too much nitrate and nitrite could contribute to the formation of N-nitroso compounds [54]. The carcinogenic effect of nitroso compounds may be promoted or enhanced by high salt intaking [4]. Additionally, high Salt intake may also promote or enhance the effect of other carcinogens [4]. (4) High salt intake increases H. pylori colonization in the stomach. H. pylori is one of the main predisposing factors for gastric cancer [4,13,14]. The cag pathogenicity island is one of the H. pylori virulence determinants, which could increase gastric cancer risk [55]. More severe gastric injury in the stomach was induced by cag-positive strains compared with cag-negative strains, and cag-positive strains further augment the risk for gastric cancer in this way [55]. Elevated salt concentrations caused an upregulation of cagA gene in some strains, enhancing cagA’s ability to translocate into gastric epithelial cells [55,56]. It indicates that high dietary salt intake could enhance the carcinogenic effects of cagA+ H. pylori strains [14]. (5) High salt intake could alter the viscosity of the protective mucous barrier, disrupt immune homeostasis and increase susceptibility to H. pylori infection [10,57,58]. These factors would result in chronic inflammation, such as atrophic gastritis and gastric ulcer, both of which are common precancerous diseases [13,14,58,59].
- Lines 30 - 35 - References 1 and 2 are not appropriate.
Response: Thank you for your advice. We have revised the references in manuscript.
We copy it here for your check:
- Mukkamalla S., A. Recio-Boiles., H.M. Babiker. Gastric Cancer. 2022.
- Sung H., Ferlay J., Siegel R. L., Laversanne M., Soerjomataram I., Jemal A., Bray F. Global Cancer Statistics 2020: GLOBOCAN Estimates of Incidence and Mortality Worldwide for 36 Cancers in 185 Countries[J]. CA Cancer J Clin, 2021,71(3):209-249.
- The authors added the recent publication from the StoP Project, however, the reference is not used in a relevant manner. Please exclude or revise.
Response: Thank you for your advice. We have excluded the reference in manuscript.
- Lines 63-64 please revise - the PRISMA statement provides reporting guidelines not “performing” guidelines.
Response: Thank you for your advice. We have revised this description throughout manuscript. We copy it here for your check:
The design, implementation, analysis, and reporting of our meta-analysis were reported in accordance with the PRISMA statement.
- Regarding: “12. How did the authors deal with duplicate publications from the same case-control study? This does not include duplicate records, which should be excluded before screening begins. Which study was selected for inclusion in the current SR and MA?
Response: Thank you for your advice. There is one point we should clarify. Study with larger sample size was chosen among duplicate publications from the same case-control study. Thank you.” — No changes have been made to the manuscript. This information must be provided to readers.
Response: Thanks to you for your good comment. According to your suggestion, we have re-written the related part. Thank you. We copy it here for your check:
2.2. Selection criteria and exclusion criteria
The studies were selected if they met all of the following criteria: (1) being a case-control study; (2) total sample size is over 100; (3) assessment of salty food intake, preference of salty food, use of table salt and relevant indexes as exposure; (4) the authors reported the odds ratio (OR) estimates, including 95% confidence intervals (CIs), for different salt intake categories. The studies were excluded if they met any of the following criteria: (1) being duplicate publications; (2) not relevant; (3) being systematic reviews, meta-analyses, meeting abstracts, letters, and dissertations without the relevant information; (4) not case-control studies; (5) OR and 95%CI not be reported. Study with larger sample size was chosen among duplicate publications from the same case-control study. Exclusion criteria were applied sequentially by screening first the titles and abstracts, and second the full text. Duplicate records were excluded before screening begins. The flow chart of the selection of studies was shown in figure 1.
- Regarding “15. The order in which articles were excluded is incorrect. The same inclusion and exclusion criteria should be applied in steps 1 and 2. Step 1 is screening based on the title and abstract, Step 2 is screening based on the full text. Duplicate records should be excluded before screening begins.” - The authors made changes to figure 1. Flow chart, however no changes have been made to the manuscript text.
Response: Thanks to you for your good comment. According to your suggestion, we have re-written the related part. Thank you. We copy it here for your check:
2.2. Selection criteria and exclusion criteria
The studies were selected if they met all of the following criteria: (1) being a case-control study; (2) total sample size is over 100; (3) assessment of salty food intake, preference of salty food, use of table salt and relevant indexes as exposure; (4) the authors reported the odds ratio (OR) estimates, including 95% confidence intervals (CIs), for different salt intake categories. The studies were excluded if they met any of the following criteria: (1) being duplicate publications; (2) not relevant; (3) being systematic reviews, meta-analyses, meeting abstracts, letters, and dissertations without the relevant information; (4) not case-control studies; (5) OR and 95%CI not be reported. Study with larger sample size was chosen among duplicate publications from the same case-control study. Exclusion criteria were applied sequentially by screening first the titles and abstracts, and second the full text. Duplicate records were excluded before screening begins. The flow chart of the selection of studies was shown in figure 1.
- Regarding: “17. Lines 124-125 - Please specify the relevant tests used to evaluate dietary salt intake. It is unclear what exposures were considered in the studies: salt taste preference, use of table salt, total sodium intake. “ - the authors have added table 2 which provides information for this, however, no changes gave been made to the manuscript text.
Response: Thank you for your advice. We have revised this description throughout manuscript. We copy it here for your check:
The study selection process and results from the literature search were shown in figure 1. Of a total of 1462 publications retrieved, 38 studies were identified that met the inclusion criteria. The relevant characteristics of the 38 studies included in the meta-analysis are reported in table 1. Overall, the meta-analysis involved 37 225 participants from 20 countries (11 studies from China, 4 from Korea, 4 from Italy, 2 from Iran, 2 from Turkey, 1 from France, England, Spain, Japan, Puerto Rico, Sweden, Mexico, Thailand, Uruguay, Colombia, Portugal, Serbia, Canada, Ecuador, and Poland). In all studies, the dietary salt intake was estimated by food frequency questionnaires or relevant tests. The estimated methods of dietary salt intake were shown in table 2. The Newcastle-Ottawa Scale was performed to assess the quality of included articles. The results of quality score were shown in table 1. A summary of characteristics and quality assessment of the included studies is listed in table 1 and table 2. The information on the adjustment variables for each study were shown in appendix table s2.
- Regarding: “18. Lines 125-126 - Please specify that the Quality assessment score is provided in Table 1.” - the authors provide the quality assessment in a supplementary table and the authors changed the manuscript text: “The Newcastle-Ottawa Scale was performed to assess the quality of included articles. ”, however, they do not state that the reader should refer to the supplementary table for these results/information. Please revise accordingly.
Response: Thank you for your advice. We have revised this description throughout manuscript. We copy it here for your check:
The study selection process and results from the literature search were shown in figure 1. Of a total of 1462 publications retrieved, 38 studies were identified that met the inclusion criteria. The relevant characteristics of the 38 studies included in the meta-analysis are reported in table 1. Overall, the meta-analysis involved 37 225 participants from 20 countries (11 studies from China, 4 from Korea, 4 from Italy, 2 from Iran, 2 from Turkey, 1 from France, England, Spain, Japan, Puerto Rico, Sweden, Mexico, Thailand, Uruguay, Colombia, Portugal, Serbia, Canada, Ecuador, and Poland). In all studies, the dietary salt intake was estimated by food frequency questionnaires or relevant tests. The estimated methods of dietary salt intake were shown in table 2. The Newcastle-Ottawa Scale was performed to assess the quality of included articles. The results of quality score were shown in table 1. A summary of characteristics and quality assessment of the included studies is listed in table 1 and table 2. The information on the adjustment variables for each study were shown in appendix table s2.
- Regarding: “20. Please provide information on the source of controls (population or hospital based). How may this impact the associations in each study?” - Please discuss the difference in the estimates (1.39 for population based controls, and 2.19 for hospital based controls). Figure 5 is not included in the manuscript (could be a supplementary figure) and results for the different controls are not provided.
Response: Thank you for your question. The detailed description has supplemented in the text. We copy it here for your check:
3.4. Subgroup analyses by regions and estimated methods of dietary salt intake
The relationship between high dietary salt intake and risk of gastric cancer was not significantly different in the geographic region, estimated methods of dietary salt intake and the source of controls (shown in figure 3, figure 4 and figure 5). The pooled ORs were changed after stratifying by geographic region. The pooled ORs of gastric cancer for the salt intake were 1.71 (95% CI, [1.51, 1.95]) for studies conducted in Europe, 1.48 (95% CI, [1.37, 1.59]) for studies conducted in Asia and 1.65 (95% CI, [1.38, 1.97]) for studies conducted in America, and there was statistically significant heterogeneity among studies of salt intake in Europe (P < 0.001 and I2 = 77.2%), Asia (P < 0.001 and I2 = 86.3%) and America (P =0.006 and I2 = 72.1%) (shown in figure 3). Furthermore, stratifying by estimated methods of dietary salt intake, the pooled ORs of gastric cancer for the salt intake were 1.38 (95% CI, [1.29, 1.49]) for studies that estimated salt addition and 2.03 (95% CI, [1.81, 2.27]) for studies that estimated consumption of salty foods or salt preference, and there was statistically significant heterogeneity among studies that estimated salt addition (P<0.001 and I2 = 87.2%) and there was statistically medium heterogeneity among studies that estimated consumption of salty foods or salt preference (P<0.001 and I2 = 66.0%) (shown in figure 4). The pooled ORs of gastric cancer for the salt intake were 1.39 (95% CI, [1.30, 1.49]) for studies that controls from community and 2.19 (95% CI, [1.93, 2.49]) for studies that controls from hospital, and there were statistically significant heterogeneity studies (I2 = 77.9% for population-based studies and I2 = 81.7% for hospital-based studies, P<0.001) (shown in figure 5).
Figure 5. Forest plot of associations between high dietary salt intake and gastric cancer risk among the source of controls.
- Regarding: “21. Please provide information on the adjustment variables for each study. A supplementary table may be necessary.” - the authors have added Table S2, however this is not ever mentioned in the manuscript text. Readers will not know this information is available unless it’s mentioned in the manuscript text.
Response: Thank you for your advice. We have revised this description throughout manuscript. We copy it here for your check:
The study selection process and results from the literature search were shown in figure 1. Of a total of 1462 publications retrieved, 38 studies were identified that met the inclusion criteria. The relevant characteristics of the 38 studies included in the meta-analysis are reported in table 1. Overall, the meta-analysis involved 37 225 participants from 20 countries (11 studies from China, 4 from Korea, 4 from Italy, 2 from Iran, 2 from Turkey, 1 from France, England, Spain, Japan, Puerto Rico, Sweden, Mexico, Thailand, Uruguay, Colombia, Portugal, Serbia, Canada, Ecuador, and Poland). In all studies, the dietary salt intake was estimated by food frequency questionnaires or relevant tests. The estimated methods of dietary salt intake were shown in table 2. The Newcastle-Ottawa Scale was performed to assess the quality of included articles. The results of quality score were shown in table 1. A summary of characteristics and quality assessment of the included studies is listed in table 1 and table 2. The information on the adjustment variables for each study were shown in appendix table s2.
- Regarding: “30. Consider discussing how the adjustment variables for each study may impact the findings of the SR and MA.” - the authors have included a supplementary table with information on adjustment variables, however they fail to mention it in the manuscript (the reader will not know it is available), and fail to discuss how adjustment variables for each study may impact their results - if studies used many different adjustment variables - this may have potentially led to increased heterogeneity. I appreciate the authors highlighting that certain risk factors were not considered as adjustment variables (Lines 89-90).
Response: Thank you for your advice. The adjustment variables were different for each study. I don’t know how to discuss impact of adjustment variables on results. Subgroup analysis or meta regression? Could you describe in detail? I will further add the relevant part in the manuscript. We have revised this description throughout manuscript. We copy it here for your check:
2.3. Data extraction and quality assessment
Two investigators (Xiaomin Wu and Liling Chen) independently conducted the literature search, reviewed the retrieved articles, and extracted detailed information from included articles.
Any disagreement about whether a study met the inclusion criteria was resolved by group discussions with the third investigator (Junxia Cheng). The following characteristics of the identified studies and respective populations were recorded: first author, year of publication, country, region, gender, age (years) (mean/range), sample size of participants, match or not, the source of controls, estimated methods of dietary salt intake, comparisons and the adjustment variables for each study. The estimated methods of dietary salt intake in the different studies were provided either in terms of total dietary salt intake or in terms of preference for salty food or both. For our analysis, we used the outcome provided for total dietary salt intake, whenever possible. Furthermore, we extracted OR estimates with the most adjustment.
- Results
3.1. Literature search and study characteristics
The study selection process and results from the literature search were shown in figure 1. Of a total of 1462 publications retrieved, 38 studies were identified that met the inclusion criteria. The relevant characteristics of the 38 studies included in the meta-analysis are reported in table 1. Overall, the meta-analysis involved 37 225 participants from 20 countries (11 studies from China, 4 from Korea, 4 from Italy, 2 from Iran, 2 from Turkey, 1 from France, England, Spain, Japan, Puerto Rico, Sweden, Mexico, Thailand, Uruguay, Colombia, Portugal, Serbia, Canada, Ecuador, and Poland). In all studies, the dietary salt intake was estimated by food frequency questionnaires or relevant tests. The estimated methods of dietary salt intake were shown in table 2. The Newcastle-Ottawa Scale was performed to assess the quality of included articles. The results of quality score were shown in table 1. A summary of characteristics and quality assessment of the included studies is listed in table 1 and table 2. The information on the adjustment variables for each study were shown in appendix table s2.
We tried our best to improve the manuscript and made some changes in the manuscript. These changes will not influence the content and framework of the paper. We appreciate the Editors/Reviewers’ warm work earnestly, and hope that the correction will meet with approval. Once again, thank you very much for your comments and suggestions.
